# EasyCreator: Empowering 4D Creation through Video Inpainting

**Yue Ma [1], Kunyu Feng [2†], Xinhua Zhang [4†], Hongyu Liu [1⋆], David Junhao Zhang [3],**
**Jinbo Xing [5], Yinhan Zhang [2], Xiangpeng Yang, Xinyu Wang [4], Zeyu Wang [2,1✉], Qifeng Chen [1✉]**

[1] HKUST, [2] HKUST(GZ), [3] NUS, [4] Tsinghua University, [5] CUHK

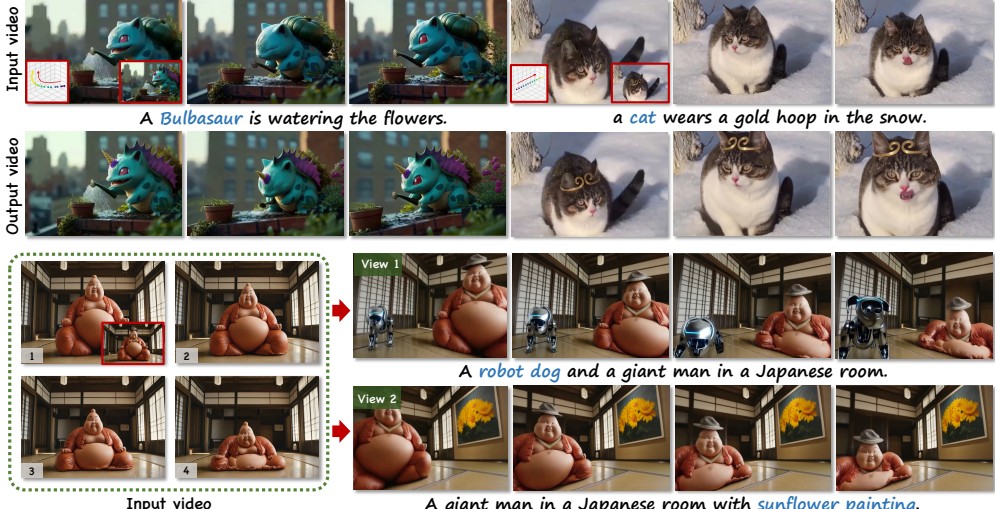

Figure 1: **Showcases of our EasyCreator**. We reformulate 4D video creation as a video inpainting task. Given an input video, EasyCreatorenables 4D video creation with various camera trajectories (bottom left on the input video's first frame) and edited first frame (bottom right), while maintaining multi-view consistency. In addition, it supports flexible prompt-based content editing (*e.g.*, adding a robot dog or sunflower painting).

## ABSTRACT

We introduce EasyCreator, a novel 4D video creation framework capable of both generating and editing 4D content from a single monocular video input. By leveraging a powerful video inpainting foundation model as a generative prior, we reformulate 4D video creation as a video inpainting task, enabling the model to fill in missing content caused by camera trajectory changes or user edits. To facilitate this, we generate composite masked inpainting video data to effectively fine-tune the model for 4D video generation. Given an input video and its associated camera trajectory, we first perform depth-based point cloud rendering to obtain invisibility masks that indicate the regions that should be completed. Simultaneously, editing masks are introduced to specify user-defined modifications, and these are combined with the invisibility masks to create a composite masks dataset. During training, we randomly sample different types of masks to construct diverse and challenging inpainting scenarios, enhancing the model's generalization and robustness in various 4D editing and generation tasks. To handle temporal consistency under large camera motion, we design a self-iterative tuning strategy that gradually increases the viewing angles during training, where the model is used to generate the next-stage training data after each fine-tuning iteration. Moreover, we introduce a temporal packaging module during inference to enhance generation quality. Our method effectively leverages the prior knowledge of the base model without degrading its

† Equal contribution
⋆ Project Leader
✉ Corresponding author

original performance, enabling the generation of 4D videos with consistent multi-view coherence. In addition, our approach supports prompt-based content editing, demonstrating strong flexibility and significantly outperforming state-of-the-art methods in both quality and versatility.

# 1 INTRODUCTION

Video generation foundation models (kel, 2024; Kong et al., 2024; Wang et al., 2025) have attracted considerable attention and witnessed rapid progress in recent years. These models are capable of synthesizing high-fidelity, temporally coherent videos from coarse user inputs such as text or image prompts. As the capabilities of such models continue to evolve, there is an increasing demand for more precise and controllable video generation (He et al., 2024; Ling et al., 2024; Wang et al., 2024a; Xing et al., 2025), where additional modalities, such as audio (Xing et al., 2024c), human pose (Hu et al., 2023; Ma et al., 2024a; Peng et al., 2024), or depth (Gen, 2023; Guo et al., 2024; Pang et al., 2024; Xing et al., 2024b), are leveraged to guide content creation. These advancements aim to better align the generated outputs with users' creative intent and enhance the expressiveness and relevance of the synthesized videos.

4D video generation is an emerging paradigm of controllable video synthesis that enables dynamic content creation guided by camera trajectories. It has garnered increasing attention for its ability to produce cinematic effects such as camera motion and bullet time, supporting immersive and expressive visual storytelling. Recent approaches (Bahmani et al., 2024; Bai et al., 2024) typically incorporate camera trajectories into pre-trained video generation foundation models by encoding them as embeddings, analogous to text prompts. These methods often rely on multi-view datasets, synthetic renderings, or monocular videos with annotated camera poses for model fine-tuning. However, despite recent progress, several limitations remain, including a strong dependence on large-scale training data, restricted input modalities, commonly limited to images or text, and limited controllability over camera viewpoints. Notably, current methods lack support for video inputs and are unable to transform user-provided monocular videos into coherent 4D representations.

To achieve more realistic generation results and enable the conversion of monocular videos into 4D representations, recent methods (Jeong et al., 2025; Ren et al., 2025; YU et al., 2025; Zhang et al., 2024a) often decompose the task into two stages. The first stage employs existing depth predictors (Hu et al., 2025) to estimate per-frame depth from monocular videos, generating dynamic point clouds that are rendered along desired camera trajectories. This rendering process typically results in videos with masked regions (holes) caused by occlusions or incomplete geometry. In the second stage, video inpainting is applied to fill these holes and produce the final output. While dynamic point clouds can now be reliably obtained using well-developed depth estimation techniques, appropriate video inpainting models for this task remain underdeveloped. Consequently, current approaches often rely on collecting additional data to fine-tune general-purpose image or text-to-video foundation models for inpainting. However, since these foundation models are not originally designed for video inpainting, they frequently fail to produce temporally consistent and visually realistic completions. Moreover, such fine-tuned models typically do not support text-based editing during generation, which significantly limits their flexibility and practicality for 4D video editing.

Fortunately, a powerful video inpainting foundation model, Wan2.1 (Wang et al., 2025) has recently emerged, trained on large-scale datasets. However, we observe that it cannot be directly applied to complete the masks (i.e., occluded regions) introduced by point cloud rendering, as such masks fall outside its training distribution. In this paper, we propose EasyCreator, a novel 4D video generation framework that reformulates 4D generation as a specialized video inpainting task. Our goal is to unlock the potential of powerful video inpainting models for 4D reconstruction, enabling realistic, flexible, and controllable results with minimal additional training. Specifically, we first utilize an off-the-shelf depth predictor (Hu et al., 2025) to estimate per-frame depth maps, which are then transformed and aggregated into dynamic point clouds. These point clouds are rendered using a double-reprojection strategy (YU et al., 2025) to generate a sequence of masks from the target camera viewpoint, projected back to the original camera poses. These masks correspond to occluded or invisible regions caused by rendering and serve as the completion targets during 4D generation. In addition, we construct an editing mask sequence to define regions requiring content modification. The occlusion and editing masks can be used individually or combined, forming a composite mask

dataset with three types of masks. During fine-tuning, we randomly sample one type of mask for each training instance, allowing the foundation model to be sufficiently trained while supporting both 4D generation and editing. Furthermore, we introduce a self-iterative tuning strategy that progressively increases viewpoint diversity by reusing results from previously trained views as training data for subsequent ones, improving stability under large camera motions.

To ensure multi-view consistency during inference, we present a temporal packing strategy that enhances coherence across frames and viewpoints by leveraging previously generated results as priors to guide content completion within masked regions. We conduct comprehensive evaluations of EasyCreator on both synchronized multi-view datasets and large-scale monocular video datasets. Quantitative results and qualitative visualizations consistently demonstrate that our method outperforms existing approaches in generating high-fidelity videos under novel camera trajectories.

## 2 RELATED WORK

**Camera-controlled video generation.** Following the success of text-to-video generation models (Bar-Tal et al., 2024; Chen et al., 2023; Ho et al., 2022; Polyak et al., 2024; Singer et al., 2023), controllable video generation with additional control signals, such as pose (Ma et al., 2024a), depth (Esser et al., 2023; Xing et al., 2024b), and sketch (Yuan et al., 2024; Meng et al., 2024; Xing et al., 2024a), has been developed to generate videos adhering to users' intentions more precisely. Camera motion control has been explored through motion LoRAs (Blattmann et al., 2023; Guo et al., 2023; Kuang et al., 2024), enabling video generation with specific camera movement patterns. For finer control, a line of work has explored employing camera conditions through intrinsic and extrinsic matrix (Wang et al., 2024b), Plucker embedding (Bahmani et al., 2024; 2025; He et al., 2024; Li et al., 2025), background-point trajectory (Feng et al., 2024; Wang et al., 2024b), point-cloud re-rendering (Yu et al., 2024), or depth-based warping (Hou et al., 2024).

**Diffusion-based video editing and inpainting.** The field of video editing has broad applications. Early studies (Cao et al., 2023; Kawar et al., 2023; Wan et al., 2024; Meng et al., 2022; Liu et al., 2025a) developed training-free or fine-tuned text-driven editing methods on images. Some works (Cong et al., 2023) extend text-to-image models, where TAV Wu et al. (2023) achieved video generation through one-shot tuning, with later works (Ceylan et al., 2023; Chai et al., 2023; Ouyang et al., 2023; Pumarola et al., 2021; Qi et al., 2023) improving temporal consistency. Video inpainting is a subtask of editing, which utilizes the user-specified mask sequences to edit the content in a video. Previous works can be classified into two categories: non-generative methods and generative methods. Non-generative methods (Hu et al., 2020; Liu et al., 2021; Zhou et al., 2023) facilitate pixel propagation using architecture priors. But they are limited to only being effective for partial object occlusions with random masks. With the development of generative models, some works (Bian et al., 2025; Zhang et al., 2023b; Zi et al., 2024) adopt the advanced text-to-video diffusion to improve their performance. They only focus on the content inpainting in video. In contrast, we reformulate 4D video creation as a video inpainting task and unlock the potential of video inpainting models for 4D reconstruction, enabling high-fidelity results with minimal additional training.

## 3 METHOD

The pipeline of our method is shown in Fig. 2. Given a video, we target the creation of dynamic 4D video content, including camera retargeting (*e.g.*, zoom, tilt, pan) and video content editing (*e.g.*, subject addition and modification). Different from previous works (Bai et al., 2025; YU et al., 2025) tuned on large-scale video datasets, we reformulate 4D generation as a specialized video inpainting task and fine-tune the model (Wang et al., 2025) to unlock its potential. In the following, we first discuss the details about the dynamic point cloud (in Sec. 3.1) and composite mask (in Sec. 3.2). Then we introduce our iterative tuning in Sec. 3.3. Finally, we present the temporal-packing inference for multi-view video consistency in Sec. 3.4.

### 3.1 DYNAMIC POINT CLOUD

Given an input video $\mathbf{V} = [\mathbf{I}_0, \ldots, \mathbf{I}_{N-1}] \in \mathbb{R}^{N \times 3 \times H \times W}$, where $N$ denotes the number of frames, our goal is to synthesize a new video sequence that follows a user-specified camera trajectory. The

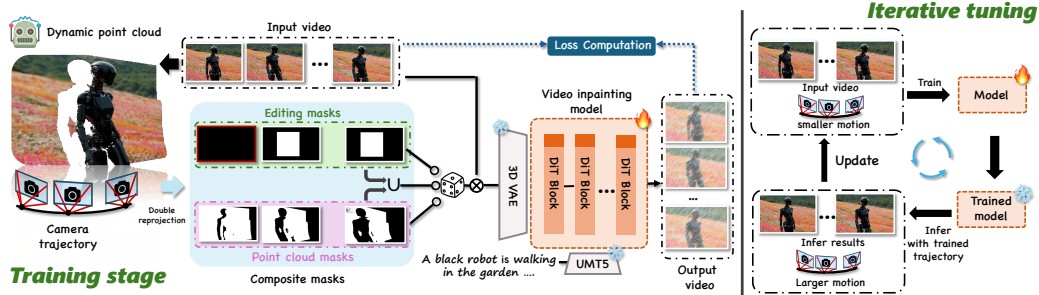

Figure 2: **Overview of our method.** We reformulate the 4D video creation as a video inpainting task. **Left:** given a video, we first generate the composite masks from the dynamic point cloud and feed them into the video inpainting model to unlock its 4D video creation capability. **Right:** To unlock the capability of generating 4D video with larger motion, we first generate videos with small motion, then feed them into the model to improve the temporal consistency progressively.

dynamic point cloud serves as a crucial intermediary representation that bridges the original frames and the novel camera views. Specifically, we use the off-the-shelf video depth estimator (Hu et al., 2025) to estimate per-frame depth map $\mathbf{D} = \{\mathbf{D}_i\}_{i=0}^{N-1}$. By combining the video frames and their depth maps, the point cloud sequence $\mathcal{P} = \{\mathbf{P}_i\}$ can be computed as follows:

$$\mathcal{P}_i = \phi([\mathbf{I}_i, \mathbf{D}_i], \mathbf{K}), \tag{1}$$

where $\phi$ is a function that maps $[\mathbf{I}_i, \mathbf{D}_i]$ to a 3D point cloud in the camera's coordinate system using $\mathbf{K}$ that represents the intrinsics of the camera described in (Chung et al., 2023; Jeong et al., 2025). Additionally, the camera motion is provided as a sequence of extrinsic matrices $\mathcal{T} = \{\mathbf{T}_1, \dots, \mathbf{T}_{N-1}\}$. With these extrinsic matrices, we can project the point cloud from each frame back to the camera plane using the perspective function $\psi$ to render an image:

$$\mathbf{I}_i^a = \psi(\mathcal{P}_i, \mathbf{K}, \mathbf{T}_i). \tag{2}$$

However, the rendered results are often incomplete, as a single monocular depth map is insufficient to reconstruct the entire scene, leading to occluded or missing regions. These missing areas can be identified during the rendering process, where a binary visibility mask $\mathbf{M} \in \mathbb{R}^{N \times 1 \times H \times W}$ is generated. Pixels with valid projections are marked as 1, while regions falling outside the original view due to camera motion are marked as 0. Our method aims to leverage a video inpainting foundation model to fill in such masked regions, thereby enabling the generation of a complete 4D video.

## 3.2 COMPOSITE MASK

Apart from the aforementioned visibility masks, our method also supports 4D editing, which requires an additional mask to indicate the user-specified editing regions. In the following sections, we describe how we construct various types of masks and integrate them into a composite mask dataset that serves as the supervision for training our video inpainting model.

**Point cloud mask.** It is hard to directly use the original visibility masks in Sec. 3.1 for supervision, since the rendered frame $\mathbf{I}_i^a$ does not contain ground-truth content for the occluded regions. To overcome this, we adopt a double reprojection strategy (YU et al., 2025) that reprojects the visibility masks back to the viewpoint of the input video. This allows us to obtain supervision masks that are spatially aligned with the original frames, and we can set the input video as the ground truth for training the inpainting model. Specifically, we back-project the rendered views $\mathbf{V}'$ into a new point cloud $\mathcal{P}'$ using $\mathbf{D}'$ and re-render the view $\mathbf{V}''$ by applying the inverse transformation $\mathcal{T}^{-1}$. This results in a paired set consisting of the corrupted video $\mathbf{V}''$ with the Mask $\mathbf{M}''$ to indicate the artifacts region and the corresponding clean video $\mathbf{V}^s$, both following the same original camera trajectory.

**Editing mask.** For 4D video generation, content editing plays a crucial role in practical applications. To leverage powerful inpainting priors and enable more flexible video editing, we adopt a content editing strategy inspired by prior work (Mou et al., 2024). During training, we randomly select a

region and generate the corresponding mask sequence. The mask for the first frame is set to '0', indicating that the first frame serves as the guidance for video synthesis. During inference, content editing is performed by modifying the first frame, and the changes are subsequently propagated to the following frames.

**Union mask.** To enable both 4D completion and 4D editing tasks simultaneously, we combine the aforementioned two types of masks using a union operation to form a new composite mask.

After obtaining various video-mask pair data, we randomly select three types of masks: point cloud masks, editing masks, and their union in the training stage. The conventional diffusion inpainting pipeline is adopt, which takes the corrupted video $\mathbf{V}''$ and the corresponding occluded mask $\mathbf{M}''$ as input conditions and predicts the completed video $\mathbf{V}^s$ using a standard flow matching loss (Wang et al., 2025).

### 3.3 SELF-ITERATIVE TUNING

By leveraging a powerful video inpainting foundation model as a generative prior, we regard the 4D video creation as a video inpainting task. However, a vanilla video inpainting diffusion model has a challenge in handling the hole video with a larger angle (*e.g.*, $> 40$ degrees). The reason behind this lies in two key factors. Firstly, in our setting, the video inpainting methods typically perform fine-tuning on a single video with various masks, which restricts their generalization ability. Secondly, current video inpainting models lack robust 3D perception capabilities. Consequently, these methods struggle to generate large-angle scene reconstructions in videos while maintaining temporal consistency across frames. To generate the video with larger-angle content, we propose the self-iterative tuning, which enhances the 3D generation ability of the video inpainting model progressively.

In detail, given a reference video $\mathbf{V} = [\mathbf{I}_0, \ldots, \mathbf{I}_{N-1}] \in \mathbb{R}^{N \times 3 \times H \times W}$ as defined in Sec. 3.1, our pipeline initiates by generating multiple video-mask pairs $\{(\mathbf{V}^{(k)}, \mathbf{M}^{(k)})\}_{k=1}^K$ using small viewpoints (*e.g.*, ¡ 30 degrees). These pairs are used to perform one-shot tuning of the video inpainting model through Low-Rank Adaptation (LoRA), and optimized parameters

$$\mathbf{W}_{LoRA}^* = \arg \min_{\mathbf{W}} \mathcal{L}\left(\mathbf{V}^k, \mathbf{M}^k, \Delta\mathbf{W}\right), \tag{3}$$

where $\Delta\mathbf{W}$ denotes the low-rank parameter updates. After tuning, we load the LoRA weight $\mathbf{W}_{LoRA}^*$ and infer the video $\widetilde{V}$ with trained small angles. Subsequent iterations employ an angle-progressive scheme: at each loop $j$, we generate a new masked video with larger angular ranges. The model then performs a self-iterative pipeline through the recurrence relation:

$$\widetilde{\mathbf{I}}_i^j = \psi\left(\mathcal{P}_i^j, \mathbf{K}, \mathbf{T}_i^j\right), \quad i = 0, 1, \ldots, N-1 \tag{4}$$

$$\widetilde{V}^j = \left\{\mathbf{I}_0^j, \ldots, \mathbf{I}_{N-1}^j\right\} \tag{5}$$

$$\mathbf{W}_{LoRA}^{(j)} = \mathbf{W}_{LoRA}^{(j-1)} + \eta \nabla_{\mathbf{W}} \mathcal{L}_{cycle}\left(\widetilde{V}^j, \mathbf{M}^j, \Delta\mathbf{W}\right) \tag{6}$$

where $\psi(\cdot)$ denotes our geometric warping function that extrapolates viewpoints. The $\eta$ is the learning rate, and $\mathcal{L}_{cycle}$ means spatial-temporal consistency MSE losses.

### 3.4 TEMPORAL-PACKING INFERENCE

After finishing the self-iterative tuning, we aim to achieve 4D video creation using various camera trajectories and the edited first frames. However, there is still a challenge in 4D video creation: multi-view video consistency. In detail, our goal is to preserve the subject and scene consistency in generated multi-view videos. Previous works, such as Recaputre (Zhang et al., 2024a), RecamMaster (Bai et al., 2025), and TrajactoryCrafter (YU et al., 2025), only focus on the consistency between the input and generated videos rather than multi-view generated videos. Reangle-a-video (Jeong et al., 2025) utilizes the image inpainting tools to improve the consistency. But it requires manual selection. In our work, during the inference stage, we propose the temporal-packing strategy to maintain the multi-view video consistency.

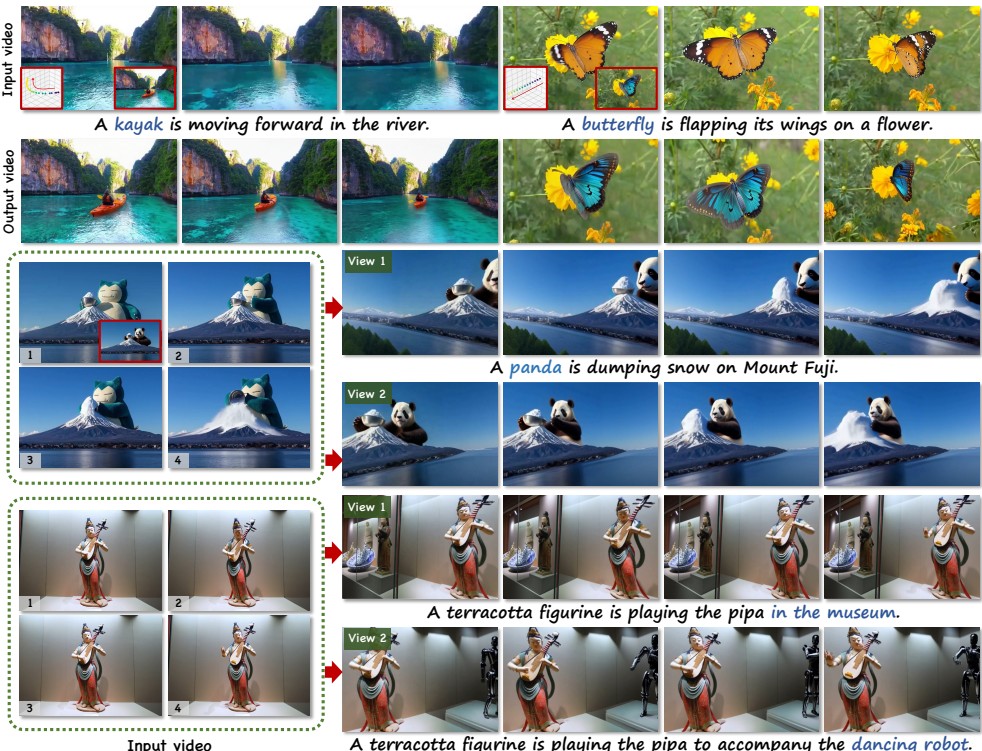

Figure 4: **Gallery of our proposed method.** Our EasyCreator enables achieving flexible and high-quality 4D video creation using the given camera trajectory and the edited first frame (2nd row). Additionally, it also supports the 4D video creation using various prompts in frozen camera ("exhibition" in 4th row and "robot" in 6th row).

Specifically, as shown in Fig. 3, we first observe that the rendered hole videos from two different camera trajectories have the overlap areas (overlap mask in Fig. 3). Inpainting the same area in two forward passes will lead to regional inconsistency. To improve the coherency of multi-view video,

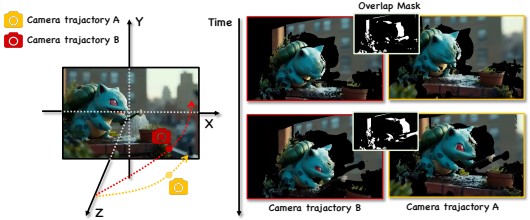

Figure 3: **Motivation of temporal-packing inference**. During the generation, there are existing overlaps (overlap mask) between various camera poses, which enables improving the consistency in multiple views.

after obtaining the generated video $\widetilde{V}^a$ from camera trajectory $\mathcal{T}^a$ using our EasyCreator, then we calculate the area of inpainting in each frame and select the $K$ frames in $\widetilde{V}^a$,

$$\mathbf{F} = \text{top-}k\text{-argmax}(\boldsymbol{S}[\widetilde{\boldsymbol{V}}^a, \mathbf{M}^a]), \quad (7)$$

where $\boldsymbol{S}(\cdot)$ notes area calculation function. In the next inference for camera trajectory $\mathcal{T}^b$, we concatenate the selected frames' tokens with the hole video token along the temporal dimension:

$$x_{input} = [\texttt{patchify}(\mathcal{E}(\mathbf{F})),$$
$$\texttt{patchify}(\mathcal{E}(\mathbf{V}^b))]_{\text{temporal}}, \quad (8)$$

where $x_{input} \in \mathbb{R}^{B \times 2N \times S \times C}$ is the input of video inpainting model, and $S = H \times W$, $C$ is the channel dimension for the latent diffusion model. $\mathcal{E}(\cdot)$ notes pretrained 3D-VAE (Kong et al., 2024). Note that we do not design any additional attention layers for feature fusion. In a pretrained video inpainting model (Wang et al., 2025), self-attention is applied globally across all tokens within the spatio-temporal attention layers.

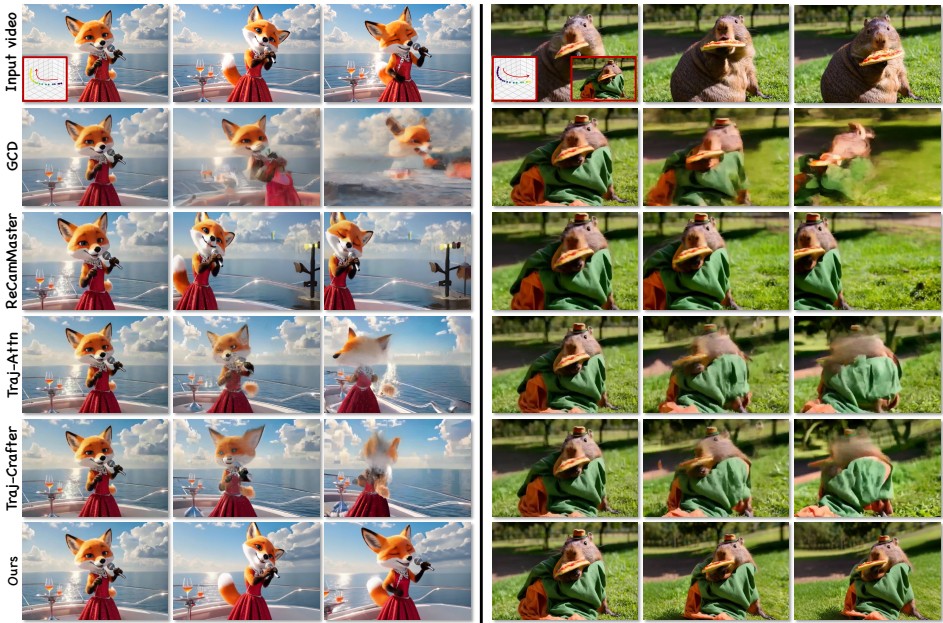

Figure 5: **Qualitative comparison results with the state-of-the-art methods.** The results show that our EasyCreator exhibits 4D video creation with better consistency and camera movements.

## 4 EXPERIMENTS

### 4.1 IMPLEMENTATION DETAILS

In our experiment, the open-sourced video generation model WAN-2.1 (Wang et al., 2025) is employed as the base text-to-video generative model. We use the LoRA (Hu et al., 2021) to finetune the model with the ranks of 128. During one-shot training, each video is input into the model as $512 \times 512$, and the video length is set to 81. The training stage is conducted for 2000 steps with learning rate $1 \times 10^{-5}$ and weight decay 0.1. For producing a dynamic point cloud, depth sequences are evaluated from input video using the open-sourced depth estimator DepthCrafter (Hu et al., 2024), with empirically configured camera intrinsics. We optimize our model using PyTorch on a single NVIDIA A800 GPU for about 2 hours. During inference, we employ the DPM solver (Lu et al., 2022) with 30 sampling steps and a text-guidance scale of 6.5. The LoRA weights are fixed at 0.7. Additional implementation details and evaluation metrics are provided in Appendix A.

### 4.2 COMPARISON WITH BASELINES

**Qualitative comparison.** We first evaluate the camera motion editing ability of our method with four baselines: Trajectory-Attention (Xiao et al., 2024b), ReCamMaster (Bai et al., 2025), TrajectoryCrafter (YU et al., 2025), GCD (Hoorick et al., 2024). The first method is a diffusion-based novel view synthesis method, and ReCamMaster and TrajectoryCrafter are state-of-the-art generative camera-retargeting methods tuned on a large-scale video dataset. GCD is a 4D novel view synthesis technique, which integrates implicit camera pose embeddings into a video diffusion model. The visual comparison is shown in Fig. 5(left). We can see that videos generated by GCD exhibit over-smoothed details and view misalignment issues. On the other hand, while TrajactoryCrafter and ReCamMaster demonstrate better pose accuracy, they struggle to produce high-fidelity frames. In contrast, our method is capable of generating novel trajectory videos with high fidelity, remarkable 4D consistency, and precise pose control.

Additionally, to test the ability of 4D video creation, we conduct a comparison with a baseline approach. In detail, we first edit the video using an advanced video editing tool (Jiang et al., 2025), then feed it into the camera-retargeting baselines for visual comparison. In Fig. 5(right), we present the visual quality of the results produced by our method and the baselines. It's observed that the 4D

Table 1: **VBench results between ours and baselines.** We collect a comprehensive video benchmark with 40 real-world videos and 40 high-quality generated videos to evaluate the performance. **Red** stands for the best result, **Blue** stands for the second best result.

| Method | VBench ↑ | | | | |
| --- | --- | --- | --- | --- | --- |
| | Subject Consis.↑ | Background Consis.↑ | Temporal Flicker.↑ | Motion Smooth.↑ | Overall Consis.↑ |
| GCD (Hoorick et al., 2024) | 0.7245 | 0.7438 | 0.6984 | 0.7041 | 0.1932 |
| Trajectory-Attention (Xiao et al., 2024b) | 0.7419 | 0.7821 | 0.7346 | 0.7528 | 0.2087 |
| ReCamMaster (Bai et al., 2025) | 0.8217 | 0.8437 | 0.8219 | 0.8523 | 0.2376 |
| TrajectoryCrafter (Mark et al., 2025) | 0.8632 | 0.8674 | 0.7925 | 0.8815 | 0.2463 |
| Ours | 0.9026 | 0.8931 | 0.8818 | 0.9242 | 0.2915 |

Table 2: **Quantitative comparison with state-of-the-art methods.** We assess visual quality, camera accuracy, and view synchronization. **Red** stands for the best result, **Blue** stands for the second best result.

| Method | Visual Quality | | | | Camera Accuracy | | View Synchronization | | |
| --- | --- | --- | --- | --- | --- | --- | --- | --- | --- |
| | FID ↓ | FVD ↓ | CLIP-T ↑ | CLIP-F ↑ | RotErr ↓ | TransErr ↓ | Mat. Pix.(K) ↑ | FVD-V ↓ | CLIP-V ↑ |
| GCD (Hoorick et al., 2024) | 73.92 | 368.44 | 32.81 | 93.66 | 2.25 | 5.78 | 638.76 | 364.28 | 85.94 |
| Trajectory-Attention (Xiao et al., 2024b) | 70.33 | 275.84 | 33.08 | 94.51 | 2.15 | 5.65 | 620.17 | 239.15 | 88.53 |
| ReCamMaster (Bai et al., 2025) | 64.82 | 162.91 | 34.68 | 96.24 | 1.48 | 5.58 | 628.45 | 153.29 | 88.27 |
| TrajectoryCrafter (YU et al., 2025) | 61.57 | 154.23 | 35.27 | 96.15 | 1.43 | 5.52 | 635.25 | 148.71 | 87.42 |
| Ours | 58.26 | 145.71 | 35.63 | 96.62 | 1.37 | 4.47 | 705.34 | 119.52 | 89.87 |

video content generated by baseline methods has significant artifacts. On the other hand, our method is capable of achieving better editing effects for 4D videos, along with smooth and accurate camera movements.

**Quantitative comparison.** We perform three comprehensive quantitative assessments of the results obtained by our proposed method and the baseline. We also apply some state-of-the-art methods to Wan-2.1 (Wang et al., 2025) and for fair comparison (See Appendix E.1). The user study is provided in the Appendix D.

(1) **Low-level metrics** in the Kubric-4D dataset (Zhang et al., 2024a). The video has a resolution of $576 \times 384$ and spans across 60 frames at 24 FPS. We select the PSNR, LPIPS, and SSIM as low-level metrics to evaluate similarity between generated and ground truth novel views. The results are reported in Tab. 3. The results clearly indicate that our method outperforms the baseline across all metrics.

Table 3: **Comparison results on Kubric-4D. Red** and **Blue** denote the best and second best results.

| Method | PSNR ↑ | SSIM ↑ | LPIPS ↓ |
| --- | --- | --- | --- |
| GCD (Hoorick et al., 2024) | 14.21 | 0.398 | 0.612 |
| Trajectory-Attention (Xiao et al., 2024b) | 14.65 | 0.426 | 0.587 |
| ReCamMaster (Bai et al., 2025) | 15.03 | 0.453 | 0.561 |
| TrajectoryCrafter (Mark et al., 2025) | 15.82 | 0.487 | 0.532 |
| Ours | 22.15 | 0.523 | 0.381 |

(2) **VBench metrics**: We collect 40 real-world videos and 40 high-quality generated videos by advanced text-to-video generative models (Kong et al., 2024; Wang et al., 2025). For each video, we generate 5 different novel trajectory videos. Five metrics in VBench (Huang et al., 2023) are employed for a more accurate evaluation (in Tab. 1)

(3) **Other metrics**: following the previous work (Bai et al., 2025), we calculate the other metrics in Tab. 2. We assess camera trajectory accuracy by calculating rotation and translation errors, following methods from earlier research in camera-guided generative approaches (He et al., 2024; Wang et al., 2024b). For view synchronization, we computed clip similarity scores and FVD between video frames from different viewpoints in the same scene, which we refer to as CLIP-V and FVD-V. Notably, EasyCreator outperforms baselines across multiple metric dimensions, demonstrating its superior generative consistency and visual fidelity.

## 4.3 ABLATION STUDY

**Effectiveness of composite mask.** To investigate the contribution of the composite mask during training, we conduct a series of ablation studies on it. The experimental settings are the same during the ablation. As shown in Fig. 6(a), when we remove the composite mask during the training stage, the results have artifacts and fail to follow the edited first frame ("French fries" on the ice). In contrast,

Table 4: **Quantitative ablation results**. Red and Blue denote the best and second best results.

| Method | Visual Quality | | | | Camera Accuracy | | View Synchronization | | |
|---|---|---|---|---|---|---|---|---|---|
| | FID ↓ | FVD ↓ | CLIP-T ↑ | CLIP-F ↑ | RotErr ↓ | TransErr ↓ | Mat. Pix.(K) ↑ | FVD-V ↓ | CLIP-V ↑ |
| W/o composite mask tuning | 78.27 | 153.28 | 30.81 | 94.21 | 1.56 | 5.26 | 518.21 | 155.47 | 85.25 |
| W/o iterative tuning | 86.29 | 197.24 | 36.58 | 92.74 | 1.49 | 4.93 | 589.29 | 204.81 | 81.26 |
| W/o temporal pack strategy | 62.46 | 168.91 | 34.92 | 93.44 | 1.51 | 4.52 | 524.63 | 137.64 | 84.71 |
| Ours | 58.26 | 145.71 | 35.63 | 96.62 | 1.37 | 4.47 | 705.34 | 119.52 | 89.87 |

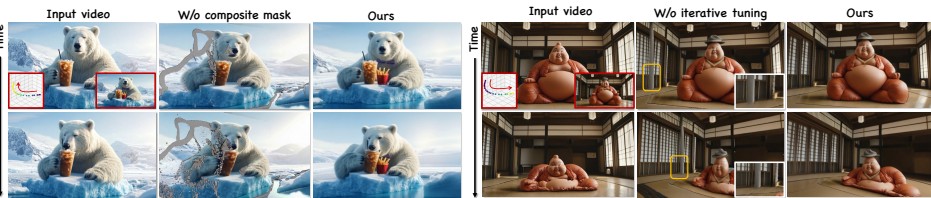

Figure 6: **Ablation study about composite mask (a) and self-iterative tuning (b).** Fig. 6 (a) demonstrates that our proposed composite mask not only keeps a smooth camera trajectory during the generation process but also enables the performance of the editing task. Fig. 6 (b) shows that the self-iterative tuning helps maintain a better temporal coherence in a larger camera motion angle.

our method enables creating a reasonable video with the subject while following the given camera trajectories. Additionally, we also perform a quantitative ablation study (in Tab. 4). Without the composite mask strategy, our method fails to achieve both the 4D video creation and editing.

**Effectiveness of self-iterative tuning.** We further assess the effectiveness of the proposed iterative tuning in Fig. 6(b) and Tab. 4. It is clearly observed that without iterative tuning, the generated video has the challenge of maintaining temporal coherence (which is marked in orange boxes ) in Fig. 6. This situation worsens when increasing the camera's motion angle. We analyze that the model lacks prior information in specific scenarios (*e.g.*, in the room), which degrades the generation capability of the video inpainting model.

**Effectiveness of temporal-packing strategy.** In Fig. 7, we show the results when there is a lack of a temporal pack strategy during inference. We first generate the video using the camera trajectory $\mathcal{T}^a$. Then we ablate the influence of the temporal pack strategy using the camera trajectory $\mathcal{T}^b$. Since there are no extra constraints and conditions, the inpainted area in multi-view video fails to preserve the consistency (which is marked in pink boxes). In contrast, our approach generates the multi-view video with consistent overlap content, which further demonstrates the effectiveness of the proposed strategy.

Figure 7: **Ablation study of temporal-packing inference.** $\mathcal{T}^a$ represents a rightward camera motion, while $\mathcal{T}^b$ turns both upward and rightward, exhibiting partial spatial overlap with $\mathcal{T}^a$. Under the temporal-packing inference strategy, our method keeps a better multi-view consistency between two trajectories.

## 5 CONCLUSION

In this paper, we present EasyCreator, a novel 4D video generation framework that reformulates 4D video creation as a video inpainting task, generating more realistic and controllable results with minimal additional training. Specifically, we first generate the composite mask using the dynamic point cloud and double-reprojection strategy. To handle temporal consistency under large camera motion, we design a self-iterative tuning strategy that gradually increases the viewing angles during training. To maintain the multi-view video consistency, the temporal-packing inference is introduced to enhance generation quality. Our method effectively leverages the prior knowledge of the video inpainting model without degrading its original performance, enabling the generation of 4D videos with consistent multi-view coherence.

**Limitations.** The limitation of our method is discussed in the appendix.

## REPRODUCIBILITY STATEMENT

All quantitative tables, qualitative images, and video results in this work are reproducible and correspond to raw model outputs without manual editing or post-hoc alteration, except for minimal format conversion and compression. After the review process, we will release a partial public repository to support reproduction, including inference scripts, example data, and example videos. The datasets, configurations, and procedures used for training and evaluation are documented in Section 4.1 and Appendix 5. We will also provide fixed configuration files and random seeds so that independent runs can reproduce the visual results within expected stochastic variation.

## ETHICS STATEMENT

Our work studies 4D video creation. To mitigate representational bias in demonstrations, we curated and display examples spanning different races, genders, and styles in the main text and appendix. All illustrative videos shown in this paper are sourced from publicly available web content; we respect the original licenses and terms of service and use the content solely for research purposes. We will not publicly release the dataset prior to completing the insertion of AI-generated watermarks and an ethics/content-safety audit. We explicitly prohibit harmful or deceptive uses of our methods and data, including deepfake attacks and other malicious generative behaviors. When any portion of our code is made public, we will enforce visible and/or machine-detectable watermarking during inference to help deter misuse. Any future releases will be accompanied by usage terms that forbid impersonation, harassment, or other malicious applications, and we will remove or restrict content that raises privacy, legal, or safety concerns.

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

APPENDIX

## A    IMPLEMENTATION DETAIL

In this section, we add more details about our framework. We select the Wan-2.1-14B (Wang et al., 2025) as our base model and fine-tune it using LoRA (14.3M). In temporal-packing inference, we select the top 10 frames as the prior. The prompts are generated by GPT4o (OpenAI, 2024). Each prompt contains about 15 words. For iterative tuning, we usually generate the corresponding rendered video and mask at intervals of 10 degrees. During inference, we need to segment the edited area using SAM2 (Kirillov et al., 2023). The first frame is edited by MagicQuill (Liu et al., 2025b). The videos in the main paper are all from the website.

## B    RELATED WORK

**Diffusion-based video editing.** Image editing is a broad and impactful field with diverse applications. Early works (Ma et al., 2026; 2025e;d;b; 2024b; Zhang et al., 2025c;b;a; Ma et al., 2025c; Feng et al., 2025; Meng et al., 2022; Yu et al., 2025) explore training-free or fine-tuning-based methods to modify image attributes via text prompts. Subsequent approaches (Zhao et al., 2024; Hui et al., 2024; Zhang et al., 2024b; Yang et al., 2024b) advance instruction-based editing by training on curated datasets. A line of research explores additional control signals, such as masked regions (Zhuang et al., 2024; Ju et al., 2024), compositing content (Song et al., 2023), customized ID with reference images (Liu et al., 2023; Kim et al., 2024), drag points (Cui et al., 2024). However, most of these works are limited to single editing tasks, making them inadequate for diverse real-world application scenarios. To address these limitations, unified frameworks (Xiao et al., 2024a; Han et al., 2024) are introduced to support various image editing and generation tasks. Recent advances in video editing can be categorized into two main approaches based on their underlying architectures. **Image-based methods** typically extend pretrained text-to-image models to the video domain. Tune-A-Video (TAV) Wu et al. (2023) pioneered this direction by adapting latent diffusion models for spatial-temporal generation through one-shot tuning. Subsequent works Qi et al. (2023); Ma et al. (2025a) improved temporal consistency through attention map fusion during inversion. Alternative approaches relying on Neural Atlas Kasten et al. (2021), dynamic NeRF deformation fields Pumarola et al. (2021); Chai et al. (2023), optical flow, Yang et al. (2023); Cong et al. (2023); Zhang et al. (2023a), feature aggregation (Jeong & Ye, 2023) significantly mitigate the temporal inconsistency issue. At the same time, they still suffer from artifacts when handling videos with large motions. **Video-based methods** leverage emerging video foundation models Yu et al. (2023); Yang et al. (2024c) to overcome some limitations of image-based approaches. Prior research efforts (Gu et al., 2023; Mou et al., 2024) demonstrate improved capabilities in motion transfer and editing by exploiting rich motion priors on single tasks. Recent works also investigate the merit of unified video generation and editing frameworks (Jiang et al., 2025).

**Novel view synthesis of dynamic scenes.** With the rapid development of diffusion models in the image and video generation domain, pre-trained video diffusion models have demonstrated strong capabilities in novel view generation. AvatarArtist (Liu et al., 2025a) leverages diffusion models to predict a 4D representation of avatars, but it is not well-suited for general scenes. DimensionX (Sun et al., 2024) integrates a dedicated motion LoRA for dynamic new view synthesis, but its camera motion is limited to a few simple trajectories. RecamMaster (Bai et al., 2025) and TrajectoryCrafter (YU et al., 2025) employ the large-scale video dataset to tune the text-to-video diffusion model, which incurs significant computational costs. Some works (Jeong et al., 2025; Zhang et al., 2024a) use masked loss and regenerate videos using point cloud rendering and mask fine-tuning along custom camera paths. Despite having LoRA adaptation capabilities, they struggle with generating 4D videos with large camera motion. In contrast, our approach can generate editable, high-quality 4D videos with multi-view consistency over a larger angle range.

## C    PRELIMINARIES

### C.1    RECTIFIED FLOW (RF)

Rectified Flow (RF) achieves high-quality generation with significantly fewer steps compared to traditional diffusion models.It establishes a linear mapping between Gaussian noise distribution and real data distributions,governed by an ordinary differential equation (ODE): $dZ_t = v(Z_t, t)\, dt, \quad t \in$

$[0, 1]$,where the velocity field $v$ is parameterized by a neural network $v_\theta$.

During training, RF defines the noise injection process through a linear interpolation:$X_t = tX_1 + (1 - t)X_0$, with the corresponding differential form: $dX_t = (X_1 - X_0) dt$.The objective is to minimize the squared error between the true derivative and the predicted velocity:

$$\min_\theta \int_0^1 \mathbb{E}\left[\|(X_1 - X_0) - v_\theta(X_t, t)\|^2\right] dt. \tag{9}$$

In the sampling process, the ODE is discretized and solved using the Euler method. Starting from a Gaussian sample $Z_{t_N} \sim \mathcal{N}(0, I)$, the model performs iterative updates:

$$Z_{t_{i-1}} = Z_{t_i} + (t_{i-1} - t_i) \cdot v_\theta(Z_{t_i}, t_i), \tag{10}$$

until the final output $Z_{t_0}$ is obtained.

## C.2 VIDEO INPAINTING DIFFUSION MODELS

Based on the latent diffusion model, we use a 3D (spatial-temporal) attention module to enable encoding and learning in the spatial and temporal dimension. This module is responsible for capturing temporal dependencies between the same spatial locations across different frames, extending the latent diffusion model into a video latent diffusion model. Therefore, the core layer of our model structure is composed of a Transformer-based diffusion model (DiT), where each DiT module consists of 3D (spatial-temporal) attention, and cross-attention modules. Specifically, we directly apply the pre-trained spatial module weights of latent diffusion models (LDMs) to initialize the spatial layers of our network, and freeze these spatial layer weights during training to maximize the inheritance of the powerful generative capabilities of LDMs. Additionally, our video generation model adopts the Rectified Flow framework for the noise scheduling and denoising process. A deterministic mapping along a straight path is used to achieve more efficient and stable generation while maintaining high-quality sample outputs.

## C.3 VIDEO DIFFUSION MODELS

Following pioneering Latent Diffusion Model (Chai et al., 2023), video diffusion models first compress the input video $V$ in pixel space into a latent space $x = \mathcal{E}(V)$ utilize a pretrained encoder $\mathcal{E}$, where the latent space $x$ can be reconstructed back to pixel space video by a decoder $\mathcal{D}$. The encoder $\mathcal{E}$ and decoder $\mathcal{D}$ are built with causal 3D convolution blocks which can encode single-frame images and multi-frame videos into the same latent space. The size of a video latent $x$ is $F \times C \times W \times H$, where $F, C, W, H$ stand for the video length, latent channels numbers, width, and height, respectively.

Recent video diffusion models (Wang et al., 2025) leverage flow matching to formulate the diffusion and denoising process in the latent space. During straining, a timestep $t \in [0, 1]$ is sampled from a logit-normal distribution, and the intermediate latent $x_t$ is defined as the linear interpolation between image or video latent $x_1$ and a random noise $x_0 \in \mathcal{N}(0, I)$ as

$$x_t = tx_1 + (1 - t)x_0. \tag{11}$$

The velocity $v_t$ is further defined as

$$v_t = \frac{dx_t}{dt} = x_1 - x_0 \tag{12}$$

The diffusion models (Wang et al., 2025; Kong et al., 2024) take intermediate latents $x_t$ as input and are trained to estimate the velocity $v_t$ using mean squared error loss.

$$\min_\theta E_{x_1, x_0 \sim N(0, I)} \|v_t - v_\theta(x_t, t, p)\|_2^2, \tag{13}$$

where $p$ is embedding the text description for the input clean video.

The inference stage starts from a Gaussian noise $x_0$, then the pretrained diffusion model gradually removes the noise in $N$ discrete timesteps $t = t_N, ..., t_0$ as $x_{t_{i-1}} = x_{t_i} + (t_{i-1} - t_i)v_\theta(x_{t_i}, t_i)$ Finally, the predicted latent $x_1$ is decoded to pixel space by the pretrained decoder $\mathcal{D}$.

Table 1: **User Study Scores**. User study scores report the average rank (1=best, 5=worst) for all the methods; lower is better. **Red** and **Blue** denote the best and second best results

| Method | Motion Pres.↓ | App.↓ | Text Align.↓ | Overall↓ |
|---|---|---|---|---|
| GCD (Hoorick et al., 2024) | 4.783 | 4.642 | 4.956 | 4.843 |
| Trajectory-Attention (Xiao et al., 2024b) | 4.283 | 4.874 | 4.642 | 4.192 |
| ReCamMaster (Bai et al., 2025) | 3.350 | 3.086 | 3.205 | 3.883 |
| TrajectoryCrafter (Mark et al., 2025) | 2.326 | **2.417** | **2.215** | 2.284 |
| Ours | **1.132** | **1.327** | **1.143** | **1.118** |

# D    USER STUDY

To achieve more comprehensive evaluation of human preferences in video quality, we perform a user study with four aspects. *Motion preservation* assesses the camera motion's adherence between given camera trajectories and generated ones. *Appearance diversity* measures the diversity according to the reference video. *Text alignment* means the semantic alignment between generated videos and prompts. *Overall* assesses the subjective quality of the generated videos. We invite 20 volunteers to provide the human feedback. The questionnaire includes 30 cases about our method and other baselines. The volunteers are asked to rank the video clips in the performance of various camera retargeting methods. (The smaller the score, the better; 1 point is the best.). Then, we calculate the average result for each baseline. The results are reported in Tab. 1. Our method present the best performance in seven metrics of objective evaluation and user study.

# E    ADDITIONAL EXPERIMENTAL RESULTS

## E.1    FAIR COMPARISON USING WAN-2.1 BACKBONE

For fair comparison, we provide the comparison with previous fitting methods in Tab. 2. ReCapture (Zhang et al., 2024a) and Reangle-A-Video (Jeong et al., 2025) are all popular fitting-based approaches, and we re-implemented them using WAN-2.1 for fair comparison. The results demonstrate that our EasyCreator consistently surpasses prior fitting methods across all quality metrics.

Table 2: **Comparison using WAN-2.1 backbone.** (re-implemented baselines). We select two SOTA approaches, ReCapture (Zhang et al., 2024a) and Reangle-A-Video (Jeong et al., 2025) for fair comparison. **Red** stands for the best result, **Blue** stands for the second best result.

| Method | Visual Quality | | | | Camera Accuracy | | View Synchronization | | |
|---|---|---|---|---|---|---|---|---|---|
| | FID ↓ | FVD ↓ | CLIP-T ↑ | CLIP-F ↑ | RotErr ↓ | TransErr ↓ | Mat. Pix.(K) ↑ | FVD-V ↓ | CLIP-V ↑ |
| ReCapture (Zhang et al., 2024a) | 65.26 | 177.45 | 33.62 | 95.41 | 1.54 | 5.62 | 633.78 | 155.29 | **88.41** |
| Reangle-A-Video (Jeong et al., 2025) | **63.19** | **171.87** | **34.08** | **95.93** | **1.46** | **5.53** | **643.51** | **151.83** | 87.62 |
| Ours | **58.26** | **145.71** | **35.63** | **96.62** | **1.37** | **4.47** | **705.35** | **119.52** | **89.87** |

## E.2    RESULTS ON REAL-WORLD VIDEOS

In Tab. 3 , we show the quantitative comparison results on real-world videos, which demonstrate that our approach achieves better performance in various metrics.

Table 3: **Quantitative comparison on real-world videos.** We assess visual quality, camera accuracy, and view synchronization. **Red** stands for the best result, **Blue** stands for the second best result.

| Method | Visual Quality | | | | Camera Accuracy | | View Synchronization | | |
|---|---|---|---|---|---|---|---|---|---|
| | FID ↓ | FVD ↓ | CLIP-T ↑ | CLIP-F ↑ | RotErr ↓ | TransErr ↓ | Mat. Pix.(K) ↑ | FVD-V ↓ | CLIP-V ↑ |
| ReCapture (Zhang et al., 2024a) | 71.28 | 280.92 | 32.15 | 93.65 | 2.21 | 5.71 | 615.31 | 244.27 | 87.61 |
| Reangle-A-Video (Jeong et al., 2025) | 70.62 | 277.46 | 32.84 | 93.87 | 2.17 | 5.68 | 616.83 | 242.91 | **87.94** |
| Trajectory-Attention (Xiao et al., 2024b) | 71.35 | 280.18 | 32.11 | 93.57 | 2.20 | 5.72 | 614.82 | 243.88 | 87.49 |
| ReCamMaster (Bai et al., 2025) | 65.78 | 167.85 | 33.73 | **95.31** | 1.54 | 5.65 | 622.68 | **158.15** | 87.32 |
| TrajectoryCrafter (YU et al., 2025) | **62.49** | **159.17** | **34.33** | 95.22 | **1.48** | **5.58** | **629.48** | 153.56 | 86.47 |
| Ours | **59.14** | **150.63** | **34.69** | **95.73** | **1.42** | **4.54** | **699.57** | **124.36** | **88.92** |

## E.3    RESULTS ON LONGER VIDEOS

To further evaluate the performance on longer videos(>30 frames), we also select 1000 samples with longer videos for comprehensive assessment. As shown in Tab. 4 , the results demonstrate that our method achieves superior performance.

Table 4: **Comparison results based on longer videos(>30 frames).** We assess visual quality, camera accuracy, and view synchronization. **Red** stands for the best result, **Blue** stands for the second best result.

| Method | Visual Quality | | | | Camera Accuracy | | View Synchronization | | |
|---|---|---|---|---|---|---|---|---|---|
| | FID ↓ | FVD ↓ | CLIP-T ↑ | CLIP-F ↑ | RotErr ↓ | TransErr ↓ | Mat. Pix.(K) ↑ | FVD-V ↓ | CLIP-V ↑ |
| ReCapture (Zhang et al., 2024a) | 85.83 | 295.84 | 13.08 | 74.51 | 4.15 | 6.65 | 600.17 | 259.15 | 73.53 |
| Reangle-A-Video (Jeong et al., 2025) | 86.12 | 296.17 | 12.87 | 74.32 | 4.42 | 7.03 | 598.62 | 258.94 | 73.26 |
| Trajectory-Attention (Xiao et al., 2024b) | 85.49 | 294.72 | 13.25 | 74.67 | 4.28 | 6.89 | 602.45 | 257.83 | **73.91** |
| ReCamMaster (Bai et al., 2025) | 80.12 | 182.91 | 19.68 | **81.24** | 3.48 | 6.58 | 608.45 | 168.29 | 73.27 |
| TrajectoryCrafter (YU et al., 2025) | **77.37** | **174.23** | **20.27** | 81.15 | **3.43** | **6.52** | **615.25** | **163.71** | 72.42 |
| Ours | **74.06** | **165.71** | **20.63** | **81.62** | **3.37** | **6.47** | **685.34** | **134.52** | **74.87** |

## E.4    RESULTS ON CHALLENGE CAMERA MOTIONS

To further evaluate the performance and robustness of our proposed model, we select 2000 samples from WebVid dataset, including many challenge camera motion videos (e.g., larger camera motions (> 50 degrees), more challenging viewpoints (backside, top-down, bottom-up), faster camera movements). In Tab. 5, our EasyCreator achieves state-of-the-art results across all quality metrics, substantially outperforming previous methods.

Table 5: **Quantitative comparison on challenge camera motion videos** (including challenging viewpoints, larger camera motions, and faster camera movements). We assess visual quality, camera accuracy, and view synchronization. **Red** stands for the best result, **Blue** stands for the second best result.

| Method | Visual Quality | | | | Camera Accuracy | | View Synchronization | | |
|---|---|---|---|---|---|---|---|---|---|
| | FID ↓ | FVD ↓ | CLIP-T ↑ | CLIP-F ↑ | RotErr ↓ | TransErr ↓ | Mat. Pix.(K) ↑ | FVD-V ↓ | CLIP-V ↑ |
| ReCapture (Zhang et al., 2024a) | 79.04 | 283.49 | 27.13 | 80.32 | 2.85 | 6.73 | 538.27 | 320.89 | 73.48 |
| Reangle-A-Video (Jeong et al., 2025) | 81.47 | 271.68 | 26.93 | 77.69 | 2.71 | 6.53 | 543.71 | 325.38 | 72.16 |
| Trajectory-Attention (Xiao et al., 2024b) | 75.26 | 312.95 | 29.17 | 80.72 | 2.68 | 6.73 | 529.47 | 276.08 | 72.53 |
| ReCamMaster (Bai et al., 2025) | 73.87 | 185.32 | 28.07 | 81.37 | 2.53 | 5.71 | 534.81 | **165.42** | **74.93** |
| TrajectoryCrafter (YU et al., 2025) | **71.26** | **174.03** | **31.15** | **81.43** | **2.36** | **5.68** | **551.74** | 169.54 | 73.01 |
| Ours | **65.32** | **168.42** | **35.17** | **88.49** | **1.98** | **5.47** | **597.71** | **136.29** | **80.36** |

## E.5    RESULTS ON MULTI-VIEW VIDEO GENERATION

We also conduct comparison experiments with multi-view video generation method to evaluate the performance of temporal-packing inference strategy, and the results are shown in Tab. 6.

Table 6: **Comparison with previous multi-view video generation approaches.** Since the CAT4D and 4Real-Video do not release the code and checkpoints. We provide the quantitative comparisons with SynCammaster (Bai et al., 2024) for multi-view comparison. **Red** stands for the best result.

| Method | Visual Quality | | | | Camera Accuracy | | View Synchronization | | |
|---|---|---|---|---|---|---|---|---|---|
| | FID ↓ | FVD ↓ | CLIP-T ↑ | CLIP-F ↑ | RotErr ↓ | TransErr ↓ | Mat. Pix.(K) ↑ | FVD-V ↓ | CLIP-V ↑ |
| SynCammaster (Bai et al., 2024) | 65.23 | 148.61 | 34.81 | 95.73 | 1.44 | 4.62 | 703.26 | 122.14 | 88.76 |
| Ours | **58.26** | **145.71** | **35.63** | **96.62** | **1.37** | **4.47** | **705.35** | **119.52** | **89.87** |

## E.6    RESULTS ON VARIOUS DEPTH ESTIMATORS

We evaluate the performance using various depth estimators in Tab. 7. The performance drops slightly compared to our original results, but remains strong. This is because our method uses the proposed mask composition strategy, which helps to better utilize the prior from the video inpainting foundation

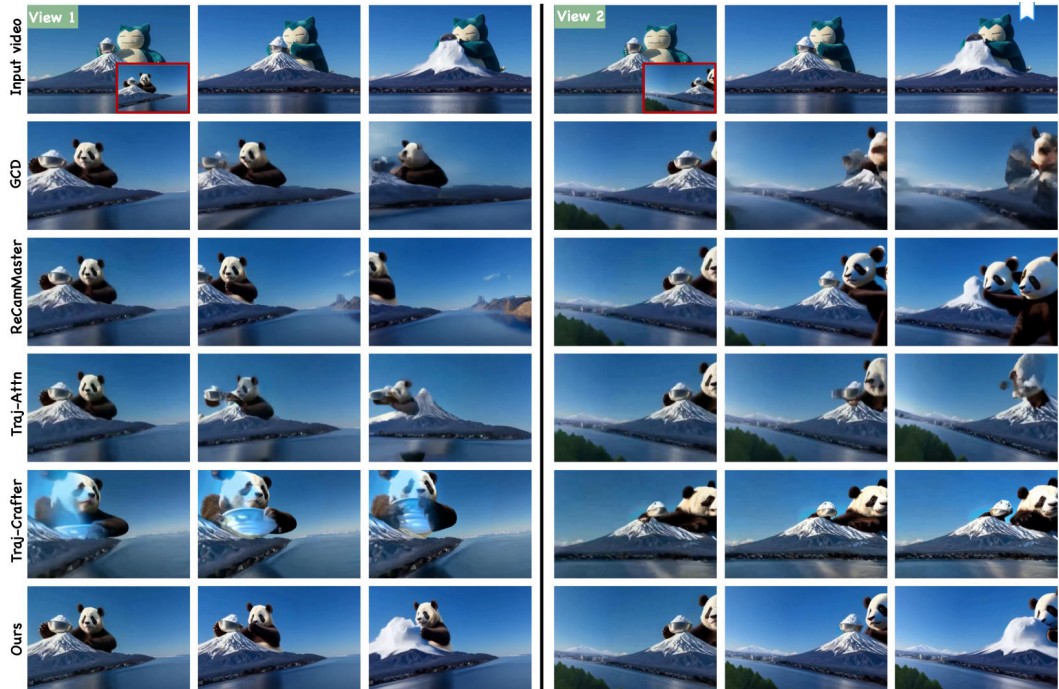

Figure 1: **More comparison results**.

model. As a result, even when the input masks are not very accurate with the erroneous depth maps, our method can still produce reasonable results. This shows that our method is robust to imperfect masks.

Table 7: **Comparison using using various depth estimators.** We select two SOTA approaches, Marigold (Ke et al., 2025) and Depth Anything (Yang et al., 2024a). **Red** stands for the best result, **Blue** stands for the second best result.

| Method | Visual Quality | | | | Camera Accuracy | | View Synchronization | | |
|---|---|---|---|---|---|---|---|---|---|
| | FID ↓ | FVD ↓ | CLIP-T ↑ | CLIP-F ↑ | RotErr ↓ | TransErr ↓ | Mat. Pix.(K) ↑ | FVD-V ↓ | CLIP-V ↑ |
| Marigold (Ke et al., 2025) | 63.78 | 173.54 | 32.45 | 93.27 | 1.98 | 6.15 | 676.21 | 142.36 | 85.92 |
| Depth Anything (Yang et al., 2024a) | 61.04 | 162.35 | 33.57 | 94.86 | 1.62 | 5.42 | 692.15 | 130.78 | 87.25 |
| Ours | 58.26 | 145.71 | 35.63 | 96.62 | 1.37 | 4.47 | 705.35 | 119.52 | 89.87 |

Table 8: **Ablation study with uncertainty estimation (UC). Red** indicates the best performance.

| Method | Visual Quality | | | | Camera Accuracy | | View Synchronization | | |
|---|---|---|---|---|---|---|---|---|---|
| | FID ↓ | FVD ↓ | CLIP-T ↑ | CLIP-F ↑ | RotErr ↓ | TransErr ↓ | Mat. Pix.(K) ↑ | FVD-V ↓ | CLIP-V ↑ |
| Ours | 58.26 | 145.71 | 35.63 | 96.62 | 1.37 | 4.47 | 705.34 | 119.52 | 89.87 |
| **Ours+UC** | **56.84** | **136.29** | **39.76** | **99.72** | **1.18** | **4.12** | **734.62** | **112.84** | **91.02** |

# F VISUALIZED COMPARISION WITH BASELINE

In Fig. 2, we provide more comparisons to evaluate the performance of the proposed method. It is apparent to discover that there is an temporal inconsistency in previous works. In contrast, our approach could effectively address the issue of temporal consistency.

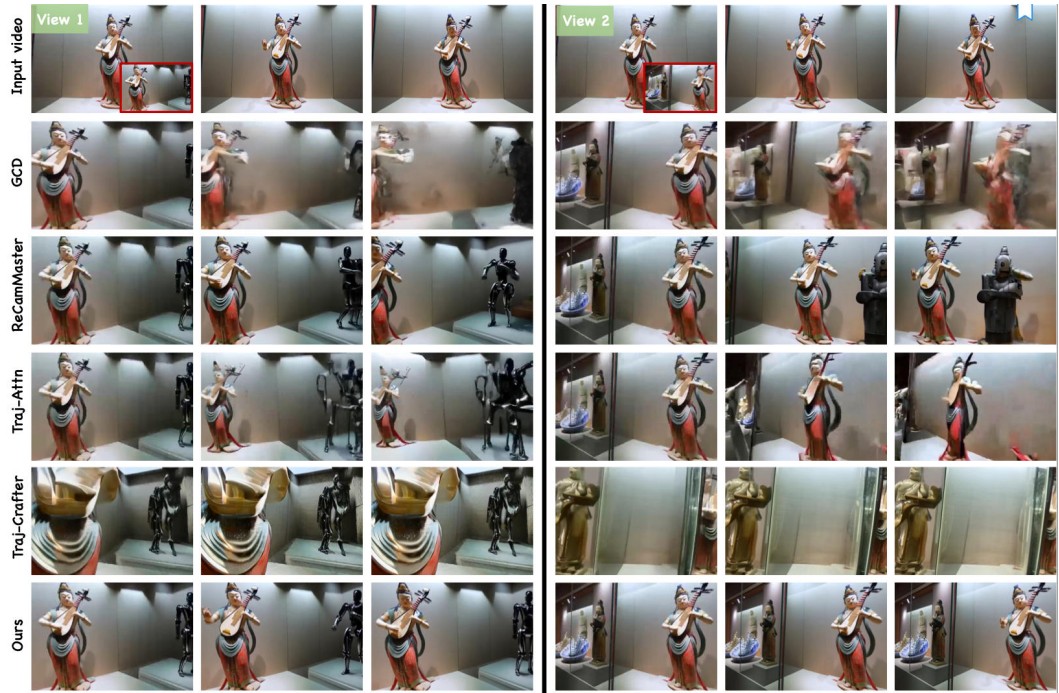

Figure 2: **More comparsions.** The prompt in view 1 is "a robot is dancing.". The prompt in view 2 is "there are a porcelain and a pottery figurine in the museum.".

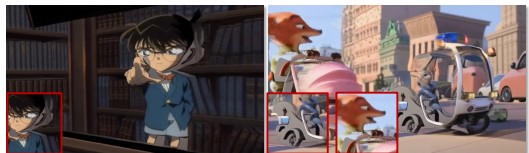

Figure 3: Limitation of our EasyCreator

# G  GALLERY OF MORE RESULTS

We show more 4D video creation results produced by our method in an MP4 file, which can be found in the file: `demo.mp4`.

# H  LIMITATION

- The limitation of EasyCreator is inherited from unreal first frame editing and inaccurate video segmentation. Since this problem relates to the base model we are using, advancements in editing and segmentation models could rectify it. Additionally, our method fails to handle free-style input video due to the capability of the pretrained foundational model, as shown in Fig. 7.

- As a fitting-based method, our method optimizes for a specific video. Compared to tuning methods, it is more time-consuming but requires fewer resources.

- Due to the use of the WAN model as the base model, which has a large number of parameters, optimizing with LoRA will take a longer time (2 hours). This issue will be improved with better models and more accelerated strategy in the future.

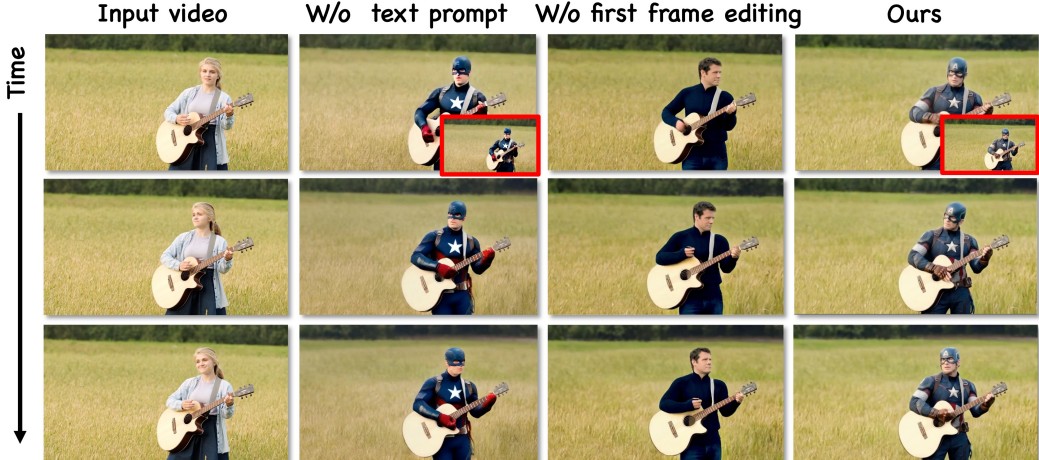

Figure 4: Ablation study about text description and first frame editing. The text prompt is "A Captain America stands in the middle of a golden wheat field, holding and playing an acoustic guitar." Using only the reference image yields significantly better results, but still falls short compared to our method, which combines both modalities.

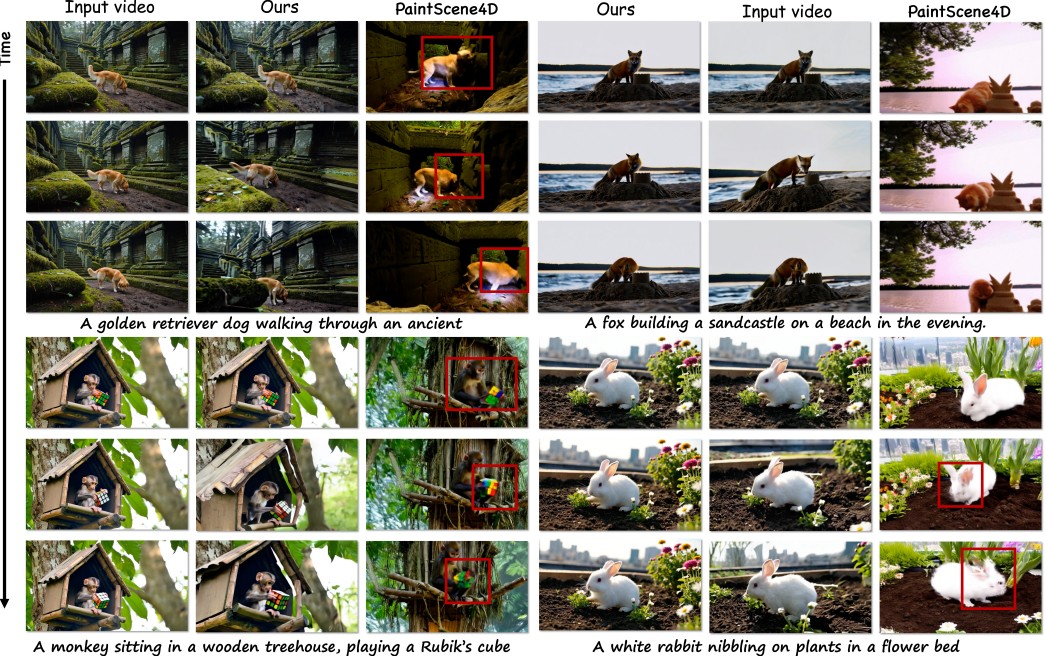

Figure 5: Comparsion with PaintScene4D. We use WAN-2.1-T2V-14B to generate the video and VIPE Huang et al. (2025) to get the camera pose.

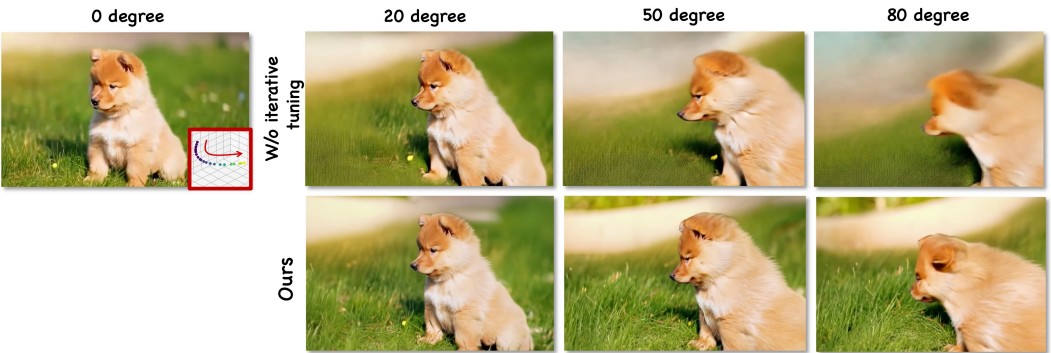

Figure 6: More ablation study about self iterative tuning. It is easy to observe that when we remove the self-iterative tuning strategy, the quality and performance of the generated results degrade a lot.

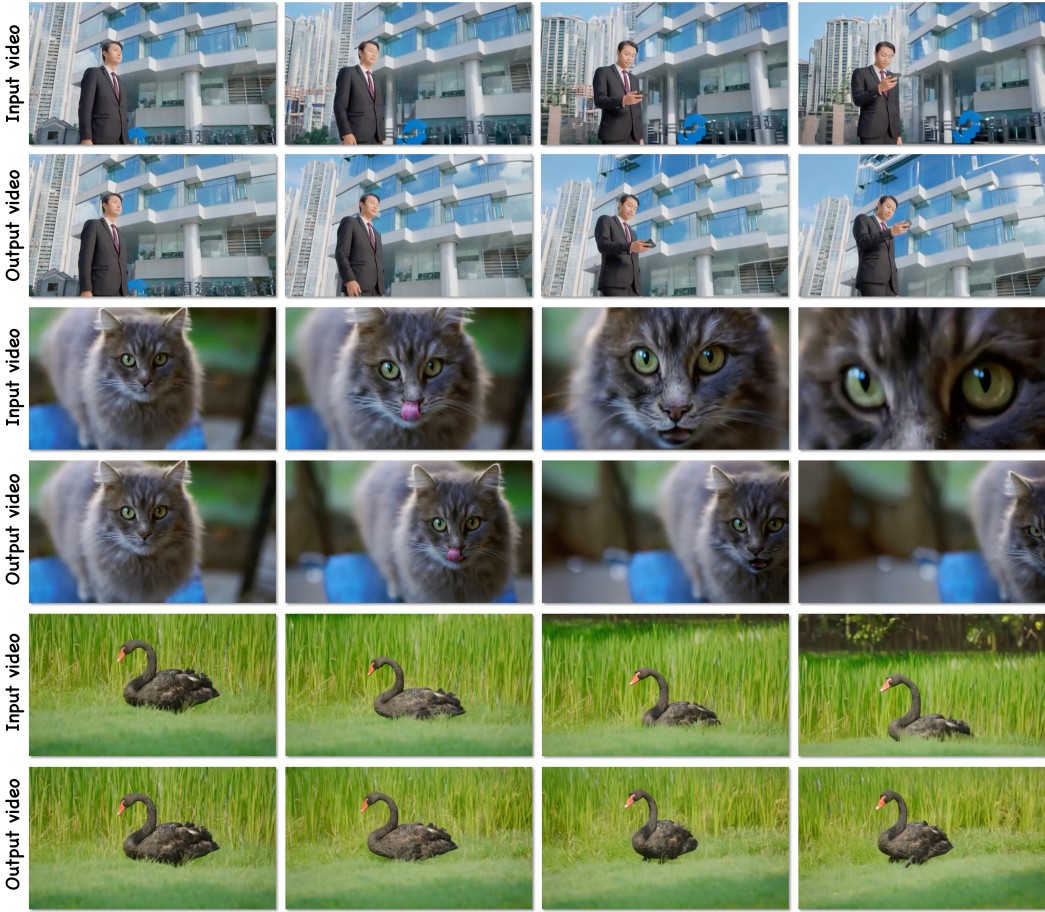

Figure 7: **More cases**. We input the videos with slight motion to evaluate our ability.

## I   SOCIAL POTENTIAL IMPACT

The introduction of EasyCreator has the potential to revolutionize the way individuals and organizations create and edit 4D video content. By enabling users to generate high-quality 4D videos from a single monocular video input, it democratizes access to advanced video production tools, making it feasible for non-experts to engage in complex multimedia projects. This can empower educators, content creators, and small businesses to produce engaging visual materials that previously required substantial resources and technical expertise.

Additionally, EasyCreator's capabilities for content editing through prompt-based interactions foster creativity and innovation, allowing users to swiftly implement changes tailored to their vision. This can lead to enhanced storytelling techniques in fields such as education, entertainment, and marketing, where immersive content captures audience attention more effectively.

From a broader perspective, the ability to generate and edit 4D videos can contribute positively to sectors like virtual reality, gaming, and remote collaboration. As these technologies become more integrated into our daily lives, they can enhance social interactions and facilitate richer experiences, ultimately promoting a more connected and informed society.

Overall, EasyCreator not only advances technological boundaries but also holds significant promise for fostering creativity, accessibility, and engagement across various fields, thereby driving social impact in the digital age.

## J   THE USAGE OF LARGE LANGUAGE MODELS

In this paper, the usage of the LLM mainly falls into the following aspects:

- **Grammar checking and format optimization**: In the paragraphs of the paper, LLMs are used for grammar error checking and format checking of charts and graphs.
- **Language polishing**: The text description part of the paper uses LLMs to polish and optimize the language expression.
- All authors are responsible for the content generated by the LLMs.

