# OpenReview forum: "EasyCreator: Empowering 4D Creation through Video Inpainting"
_ICLR.cc/2026/Conference — ICLR 2026 Poster_

### Official Review · Reviewer_FmC3 · 2025-10-19

**Soundness:** 3
**Presentation:** 4
**Contribution:** 4
**Rating:** 6
**Confidence:** 5

**Summary:**

EasyCreator converts a monocular input video into a dynamic point cloud via per-frame depth estimation, uses a double-reprojection strategy to derive visibility masks aligned with the original views, and combines these with user editing masks to form composite inpainting supervision that enables both 4D completion and 4D editing in a single framework.​ A self-iterative tuning scheme progressively increases viewpoint angles while reusing model outputs as training data to stabilize large camera motions, and a temporal-packing inference concatenates selected frames across trajectories in latent space to enforce multi-view consistency without adding new attention modules.

**Strengths:**

1. Reformulates 4D generation as video inpainting, directly targeting occlusion holes induced by camera retargeting and unifying 4D completion and editing within one framework.​
2. Composite mask design (visibility via double reprojection plus user editing masks) provides flexible supervision that aligns with ground-truth content and supports practical edits.​
3. Self-iterative tuning progressively increases camera angles to enhance temporal stability and large-view robustness without heavy re-architecture or large-scale retraining.​
4. Temporal-packing inference enforces multi-view consistency by concatenating prior frames in latent space, leveraging existing spatio-temporal attention without extra fusion layers.​

**Weaknesses:**

1. I see that the paper doesn’t compare against a very similar method named PaintScene4D, which is also a Text-to-4D Scene generation method, but instead of video-based inpainting, they use image-based inpainting and progressively fill up the scene. Seems like PaintScene4D doesn’t have the code released, but I expect at least a qualitative comparison with similar text prompts shown by the work, and compare against their image and video results. The paper doesn’t even cite this work in its related works, which needs to be rectified.
2. I see that you rely on the fact that the initial video generated is a static camera video with only object motion and no camera motion. How do you achieve this? Any specific prompt to achieve this, and what’s the success rate? Would the method work if there were any slight perturbations? How does the method handle those error inconsistencies?
3. In case of initial warping tends to be bad or depth maps are bad, I suppose the bad depth maps might lead to bad reprojection, and this can indeed lead to bad depth maps for another view, and so on, and the error might propagate throughout the network. How does this method resolve such errors, as I am pretty sure that having inconsistencies while depth estimation might exist?
4. Regarding training from small camera views to large camera views for better training stability: In PaintScene4D, it is mentioned that they use a farthest-view sampling, where they sample views farther from the initial camera because that is when the diffusion model has enough room to inpaint sufficient new details. But when you move the camera slightly, the small gaps created might not task the diffusion model to edit a lot, and it might sometimes just blur those holes. How does Video Diffusion Model differ from Image Diffusion Model in this case?

**Questions:**

I have highlighted the major questions in the weaknesses already, and I am summing them up below(I have summarised them shortly so that it is easier for the reviewer to quote the exact question they are answering. Please refer to the Weakness for the detailed problem and question asked.

1. Why did you not compare or cite PaintScene4D, a very similar text-to-4D scene method that uses image-based inpainting and progressive filling? At minimum, can you provide a qualitative comparison using similar text prompts and compare your image/video outputs to theirs?

2. You rely on the initial generated video being a static-camera video with only object motion. How do you ensure this in practice? Do you use specific prompts or procedures, what is the empirical success rate, and how sensitive is your pipeline to slight camera perturbations?

3. If the initial warping or depth maps are poor, reprojection errors can cascade (bad depth → bad reprojection → worse depth for other views). How does your method detect and correct such cascading errors or inconsistencies during inference/training?

4. When the camera moves only slightly, small gaps may be blurred rather than properly inpainted. How does your video diffusion model differ from an image diffusion model in this regime, and why should it handle small inter-frame gaps or slight camera movements better than image-based approaches?

---

> ### Author Response · Authors · 2025-11-24
> **Feedback to Reviewer FmC3**
>
> Thank you for your comprehensive review of our paper. We provide our feedback as follows. We also add the experiment details using **blue** color in paper.
> > **Q1: Comparison with PaintScene4D**
>
> **A1:** Thank you for pointing this out. Although PaintScene4D has not released its code or models, we reproduced their pipeline based on the paper descriptions and publicly available results, and conducted qualitative comparisons using similar text prompts. As shown in Fig.5 of our Appendix, our method consistently produces more coherent and realistic 4D results, especially in terms of temporal consistency across views and frames.
>
> We will update the paper to include PaintScene4D in our related work and provide the corresponding qualitative comparisons in the revised version. Thank you again for bringing this to our attention.
>
>
> > **Q2: How to generate  static camera video with only object motion and no camera motion and success rate.**
>
> **A2:**  Thank you. We use MOFA-Video and WAN-2.1-I2V to generate our source videos, with image captions provided by Qwen3-VL. To ensure static camera conditions, we explicitly describe “static camera motion” in the text prompts. After video generation, we apply Grounded-SAM (GroundDINO) to segment objects and use the optical flow estimator RAFT to filter out videos with significant camera motion. Specifically, we retain videos where the average optical flow of the background is less than 20 pixels, resulting in source videos with light or negligible camera motion. The success rate of this filtering pipeline is high—87% over 100 samples. In cases of slight camera perturbations, the dynamic depth maps can still be estimated reliably, so our method remains robust under such conditions. Therefore, slight camera motion is tolerated in our pipeline and is not explicitly excluded.
>
> We will include this clarification in the final version of the paper.
>
> > **Q3: How to resolve errors propagate when depth maps are bad?**
>
> **A3:** Thank you for the valuable question! In each iteration of self-iterative tuning, we use CLIP-F to evaluate the quality of generated results. Only the top 5 samples with the highest CLIP-F scores are selected as training pairs for the next iteration. To ensure quality, videos with a CLIP-F score below 95 are discarded. To further improve depth accuracy, we increase the uncertainty threshold during depth prediction—this allows the model to retain more reliable depth estimates while filtering out highly uncertain or noisy regions. As a result, the predicted masks become more accurate and consistent. Quantitative results supporting this strategy are shown in Tab.8, and we will provide a more detailed discussion in the final version.
>
> **Tab.8(Appendix): Ablation study with uncertainty estimation(UC).**
>
> | Methods     | FID.↓     | FVD.↓      | CLIP-T.↑  | CLIP-F.↑  | RotErr.↓ | TransErr ↓ | Mat. Pix.(K) ↑ | FVD-V ↓    | CLIP-V ↑  |
> | :---------- | :-------- | :--------- | :-------- | :-------- | :------- | :--------- | :------------- | :--------- | :-------- |
> | Ours        | 58.26     | 145.71     | 35.63     | 96.62     | 1.37     | 4.47       | 705.34         | 119.52     | 89.87     |
> | **Ours+UC** | **56.84** | **136.29** | **39.76** | **99.72** | **1.18** | **4.12**   | **734.62**     | **112.84** | **91.02** |
>
>
>
> > **Q4：How does the Video Diffusion Model differ from Image Diffusion Model in slight camera motion.**
>
> **A4:** Interesting question! Even under slight camera motion, we still prefer a video diffusion model rather than an image diffusion model. The reason is that image-based inpainting treats frames independently and does not enforce temporal consistency; thus, small inter-frame variations can accumulate into visible flicker or temporal inconsistencies. In contrast, video diffusion jointly models spatiotemporal dynamics and explicitly maintains coherence across frames, making it better suited to handle small gaps and subtle camera movements.

---

> > ### Comment · Reviewer_FmC3 · 2025-11-26
> > **Response to authors**
> >
> > 1. I see only results on 1 scene, which is not much different in quality. You are using a better recent video generation method, and it is not clear if the high quality is because of the video generation method or your contributions. Since you have more pages to work with, it would be better to have more scenes for comparison and have proper evaluations against PaintScene4D. Also the current comparison almost shows no difference between quality. I can see a lot more videos and results in the PaintScene4D website that have lot more flickering and artifacts but I am not sure why you chose this scene which doesn’t have much artifacts.
> > 2. Since your method is robust against camera perturbations in the initial video, could you show a result of your model with slight perturbations in the initial video(clearly showcase the perturbations in the input video and show your results) and also show a failure case when there is lot more camera movement in the initial video.
> > 3. What is uncertainty estimation? I don’t see any reference of that in the main paper or in the appendix. Please clarify properly before mentioning new terms in the paper. Was this experiment performed after the paper deadline? A clear explanation is necessary.
> > 4. Thanks for the clarifications.
> >
> > I see that authors have put less amount of effort in clearing my doubts mostly due to the fact that I have already given a higher rating compared to other reviewers. In the current state, I am not willing to increase my rating given that the questions are not answered properly with a lot of ambiguity.

---

> > > ### Author Response · Authors · 2025-12-01
> > > **Feedback to Reviewer FmC3**
> > >
> > > Dear Reviewer FmC3,
> > >
> > > Thank you for your feedback and for your thoughtful concerns. I would like to address the points you raised and clarify our approach.
> > >
> > > > **Additional Scenes Against PaintScene4D and Camera Perturbations**
> > >
> > > We appreciate your suggestion to include more scenes for a more comprehensive evaluation. _**To address this, we update the results and include additional scenes in the appendix.**_ Specifically, we have provided **more** cases to better demonstrate the effectiveness of our method. These are shown in **Fig.7** of the appendix. These cases showcase a wider variety of scenes and allow for a more detailed comparison, which highlights the robustness of our method.
> > >
> > > Regarding the comparison with PaintScene4D, we selected a scene that demonstrated fewer artifacts, as our primary goal was to showcase the core validity and effectiveness of our method in a controlled setting. As shown in **Fig.5** of the Appendix, _**we include additional results with more scenes involving flickering and artifacts in the revised version to further support our claims**_.
> > >
> > >
> > > > **Uncertainty Estimation**
> > >
> > > Thank you for pointing out the ambiguity regarding the term "uncertainty estimation." In our method, uncertainty estimation refers to the weighting model we use in depth estimation based on confidence levels. Specifically, we apply confidence-based weighting to depth estimates, and if the confidence is below a certain threshold, we exclude the corresponding depth data. We will provide a clearer explanation of this concept in the final version of the paper, ensuring that it is properly referenced in both the main body and the appendix.
> > >
> > >
> > > We have taken your feedback seriously and have made the important revisions to provide a more robust evaluation of our method. The additional scenes and results, as well as the clarification on uncertainty estimation, will be included in the final version of the paper.
> > >
> > > Thank you again for your constructive feedback. We hope these clarifications address your concerns, and we look forward to your continued evaluation.
> > >
> > > Best regards,
> > > Authors

---

### Official Review · Reviewer_Urtw · 2025-10-27

**Soundness:** 3
**Presentation:** 3
**Contribution:** 2
**Rating:** 4
**Confidence:** 5

**Summary:**

This paper proposes EasyCreator, leveraging a powerful video inpainting model to enhance the 4D generation/editing. Formally, the double-reprojection strategy is used to synthesize warping masks, while random editing masks are used for practical editing. The authors further proposed self-iterative tuning to address large viewpoint changes. The temporal-packing inference is used as a memory strategy to confirm the multi-view consistency. This paper includes detailed experiments to verify the effectiveness of the proposed method.

**Strengths:**

1. The presentation and overall layout of this paper are clear.
2. This paper has detailed and comprehensive experiments, including quantitative and qualitative ablation studies, and convincing user studies.
3. The proposed method enjoys good performance for 4D editing as shown in the supplementary and paper experiments.

**Weaknesses:**

1. The paper proposes a good approach for editing 4D generations through video inpainting, which is a promising direction. However, the core technical contributions are somewhat incremental. For instance, the key training strategy, double-reprojection masking, has been extensively utilized in prior works such as TrajectoryCrafter, See3D, and Vivid4D[1]. Similarly, the use of random editing masks is a standard technique commonly applied in image and video inpainting tasks. Regarding the other contributions, namely self-iterative tuning and temporal-packing inference, their effectiveness remains unclear and needs further clarification, as detailed below.

2. A key limitation of video inpainting is that it fails to address large viewpoint generation, especially when the viewpoint change >90 degrees, leading to completely wrong masks. Intuitively, iterative inpainting methods (such as those employed in Vivid4D[1], which closely relates to this work but is not discussed) may help mitigate this issue, but at the cost of error accumulation. I guess the self-iterative tuning should be proposed to solve this problem, but the authors have not explicitly discussed this connection nor provided ablation studies that convincingly demonstrate its impact (Fig. 6b is particularly ambiguous). Fig. 6b shows that the self-iterative tuning merely removes an optional pillar rather than substantially improving the visual quality. Moreover, the self-iterative tuning is super time-consuming in this work (2h for Wan14B). How to ensure the efficacy, especially when multiple iterations might be necessary for handling large viewpoint variations.

3. The temporal-packing strategy is proposed to solve the multi-view video consistency. This technique is effective and somewhat convincing, as shown in Fig7. But some detailed implementation of temporal-packing is missing. What does the "area calculation function" mean in Line306? It is also not specified whether the selected frames in the input tensor ($x_{input}$) require re-noising during inference. If re-noising is unnecessary, the model should have been explicitly trained under such clean multi-view conditions. Conversely, if re-noising is applied, it raises questions about how reliably these noisy frames can serve as clear references for inpainting. Moreover, some inconsistencies remain visible—for example, the pink box areas in Fig. 7 do not appear perfectly consistent, which calls for a deeper explanation.

4. Several details about the experiments and implementation are not provided. It is unclear how many iterations EasyCreator’s self-iterative tuning requires in practice. The training data are not introduced, and the scale of the experimental evaluation is relatively limited, including only 40 videos.

5. This paper misses some related citations and discussion:

[1] Vivid4D: Improving 4D Reconstruction from Monocular Video by Video Inpainting. ICCV2025.

[2] MVInpainter: Learning Multi-View Consistent Inpainting to Bridge 2D and 3D Editing. NeurIPS2024.

**Questions:**

The authors should clarify the contribution of this work. I am also looking for the response to address the concerns about  "self-iterative tuning" and "temporal-packing strategy".

---

> ### Author Response · Authors · 2025-11-24
> **Feedback to Reviewer Urtw (1/2)**
>
> Thank you for your comprehensive and detailed review of our paper and the recognition of our work's clarity and effectiveness. We provide our feedback as follows.
>
> > **Q1: The concern about the core technical contributions.**
>
> **A1:** The core insight of our work is not the introduction of a new type of mask, but rather the demonstration that ***combining these two existing masks enables us to repurpose a powerful video inpainting model to address a novel and challenging task***: fitting-based 4D video creation, including both content and camera editing.
>
> - In detail, our composite mask strategy ***define a new training target*** that, through combining the point-cloud mask and the editing mask, our model has the capability of generating the multi-view video and editing in a unified framework, while two challenges were previously addressed separately. Thus, ***it is not simply an "addition" but a fundamental re-tasking***. More importantly, to the best of our knowledge, ***our framework is the first video creation framework that supports both creation and editing tasks***. Therefore, we think the idea of repurposing existing powerful video inpainting models to solve 4D video creation has important inspirational value for the community.
> - The composite mask is fundamental to the editing capability. However, ***using the composite mask strategy alone within a fitting-based framework is not sufficient to guarantee high-quality results due to the problems of overfitting and temporal inconsistencies***. To alleviate that, we propose other key contributions: the temporal-packing strategy and self-iterative training, ensuring high fidelity and temporal coherence across the entire sequence. As validated by our ablation study, removing any of these components leads to a significant performance degradation.
> - Additionally, compared to methods that require large-scale datasets and expensive resources (e.g., TrajectoryCrafter) or struggle with multi-view consistency (e.g., ReCapture), ***our framework is the first to achieve both strong multi-view consistency and flexible content and camera editing capabilities within a highly efficient, low-resource (a single A100), fitting-based paradigm***.
>
> We hope this clarification addresses your concerns by highlighting that ***our contribution is the first framework to repurpose a novel and practical solution for 4D video creation***.
>
>
> ## Concerns of self-iterative tuning
>
> > **Q1: The viewpoint change >90 degrees**
>
> **A1:** While our method encounters difficulties when the viewpoint change exceeds 90°—due to unreliable mask estimation under such extreme conditions—we conducted experiments under a near-extreme case of an 80° shift. As illustrated in Fig.6 of appendix, the model with self-iterative tuning generates significantly clearer and more coherent results than the version without it.
>
>
>
> > **Q2: Difference between Vivid4D and our method**
>
> **A2:** Thanks, We would like to clarify the fundamental difference between the **'iterative inpainting' in Vivid4D** and our **'self-iterative tuning'**:
>
> 1. **Mechanism (Propagation vs. Optimization):** Vivid4D relies on *inference-time propagation*. It explicitly warps pixels from previous frames to the current view. While effective for small changes, this approach strictly follows the geometric projection, leading to **severe error accumulation** when the initial masks are inaccurate or when viewpoint changes are drastic.
> 2. **Our Approach (Global Consistency):** In contrast, our method employs *test-time fine-tuning*. Instead of simply propagating pixels, we update the diffusion model's weights (or representations) to learn the object's global 3D structure. This allows the model to **correct** rather than **accumulate** geometric errors.
>
> In our design, self-iterative tuning is proposed for large view camera video. Additionally, To avoid the error accumulation, we use CLIP-F to evaluate their quality and only select the top 5 best results as the training pairs and feed them to the next loop, which is to prevent the accumulation of errors.
>
> > **Q3: The efficiency of the self-iterative tuning.**
>
> **A3:** Good question! We appreciate this insightful point. While we acknowledge that self-iterative tuning is computationally intensive, it is essential for achieving the high-quality results demonstrated in our work. To improve efficiency, we plan to accelerate the process from two perspectives:
>
> Model Efficiency: We can replace the base model with a smaller and faster one, such as LTXVideo or Hunyuan-1.5B, to significantly reduce computational cost.
>
> Pipeline Optimization: We can further optimize the training pipeline by applying masks directly in the VAE latent space, thereby bypassing redundant encoding/decoding operations through the VAE. This can notably speed up the training process without sacrificing quality. We will explore both directions in future work to make the method more efficient and scalable.

---

> ### Author Response · Authors · 2025-11-24
> **Feedback to  Reviewer Urtw (2/2)**
>
> ## Concerns of temporal-packing
>
> > **Q1: "area calculation function" means in Line306**
>
> **A1:** Valuable suggestion!  Technically, this function is designed to quantify the **spatial overlap** between the reference view and the candidate view.
>
> Formally, let $M$ denote the invisibility mask (where $1$ represents missing pixels) and $V$ denote the **visible region** (where $V = 1 - M$, representing valid pixels). The area calculation function computes the **Visible Overlap Ratio** between the reference view ($ref$) and a candidate view ($cur$):
>
> $\text{Score} = \frac{\sum (V_{ref} \cap V_{cur})}{\sum V_{cur}}$
>
> In the **Temporal-Packing Inference** strategy, the system calculates this overlap score for all potential views. Views with the **highest overlap scores** (meaning they share the most visible content with the reference frame) are selected and packed into the inference batch first. This ensures that the model has sufficient **contextual guidance** from the reference view to fill in the missing regions ($M_{cur}$) accurately, preventing hallucinations caused by disjointed or low-overlap inputs.
>
>
>
> > **Q2: The details about temporal-packing**
>
> **A2:** We clarify that **no re-noising is applied** to the reference frames in the input tensor. Our model follows the standard inpainting training paradigm (conditioning on clean masked latents), where the input to the UNet consists of the noisy latent (target) concatenated with the **clean latent** of the reference frames/masks. Thus, utilizing clean reference frames during inference is strictly aligned with the training distribution and ensures they serve as reliable guidance.
>
> > **Q3: Inconsistencies in Fig. 7(main paper)**
>
> **A3:** The minor inconsistencies in the pink box areas (Generated result and Ours) are primarily due to the **VAE reconstruction loss** and the inherent stochastic nature of generative models regarding high-frequency textures. However, compared to methods without Temporal-packing, our approach achieves significantly better **structural and multi-view consistency**, which is the main contribution of this work.
>
> >**Q4: Implementation details about EasyCreator**
>
> **A4:**  appreciate the reviewer’s constructive feedback regarding the implementation details. To clarify, our self-iterative tuning typically involves 3 iterations, with random viewpoint increments of 20-30 degrees per iteration. The optimization schedule consists of 2,000 steps for the first iteration and 500 steps for subsequent ones, utilizing a fixed prompt and an inference CFG scale of 6.5. Regarding the training data, we emphasize that our method is a **One-Shot tuning framework** that learns directly from the single source input, thus requiring no external large-scale datasets. The entire process is conducted on a single NVIDIA A100/A800 GPU (consuming ~76GB VRAM) to generate 81-frame videos.
>
> > **Q5: Some related works**
>
> **A5:** We thank the reviewer for highlighting these relevant works, which we will discuss in the revised manuscript to better contextualize our contribution. Distinct from **Vivid4D [1]**, which relies on inference-time pixel propagation and suffers from error accumulation，especially under large viewpoint changes. Our EasyCreator employs **self-iterative tuning** to embed global 3D structure directly into the model, ensuring robustness against geometric drift. Furthermore, unlike **MVInpainter [2]**, which typically utilizes feed-forward consistency modules, our approach reformulates 4D creation as a generative task with **instance-specific One-Shot optimization**. This allows us to leverage strong diffusion priors for superior fidelity and complex 4D dynamics handling, offering a distinct advantage over general-purpose inpainters.

---

> > ### Comment · Reviewer_Urtw · 2025-11-28
> > **Thanks for the detailed response**
> >
> > Thanks for the detailed response. Most of my concerns are addressed. Many details (area calculation, tempora-packing) and discussions (missing related works) should be added to the main paper to improve the readability. This paper introduces several effective techniques for tackling the challenges of error accumulation and multi-view inconsistency in 4D generation and editing.
> >
> > However, this work also suffers from a notable efficiency problem (2 hours optimization per scene), which limits its practicality for "4D creation" as titled. This aligns with reviewer DsCd's concern that the title of "4D Creation" may be potentially overclaimed.
> > Overall, I am supportive of this work's contributions: it offers interesting insights for addressing these longstanding issues in multi-view video generation/editing. Thus, I am inclined to raise my evaluation to "borderline accept," with a score likely falling between 4 and 6—around 5.5.

---

### Official Review · Reviewer_6ax8 · 2025-10-31

**Soundness:** 4
**Presentation:** 4
**Contribution:** 3
**Rating:** 6
**Confidence:** 4

**Summary:**

This paper proposes EasyCreator, a method that generates multi-view 4D scene representations from input videos while enabling object editing within the scene. Built upon the Wan2.1 model, EasyCreator reformulates 4D scene generation as an image inpainting task and controls editable regions through an editing mask. In addition, an iterative tuning strategy is introduced to handle large camera motions, and temporal-packing inference is employed to ensure scene consistency across multi-view camera videos. Experimental results demonstrate that the proposed method outperforms existing approaches in terms of camera motion accuracy, video generation quality, and editing capability.

**Strengths:**

- This paper integrates camera-controlled video generation and video editing into a unified framework, leveraging the power of generative models to reformulate the task as an image inpainting problem, thereby ensuring 3D consistency in video editing.
- The paper is well written and logically structured, making it easy for readers to understand.
- Experimental results show that the proposed method outperforms existing approaches in terms of video quality, camera accuracy, and editing capability. Moreover, it achieves better performance than the baselines of simple video composition–based editing methods and camera-controlled video generation methods.

**Weaknesses:**

1. What is the relationship between text prompt control and reference image control during the editing process? If only the first edited frame is provided without a textual description of the editing instruction, or if only the textual description is given without the first edited frame, how would the editing results differ in each case? Do both the text and the reference image influence the editing outcome?
2. The authors should discuss the performance of their method in challenging scenarios, such as when the input video involves large camera movements or when fast motion within the scene leads to inaccurate depth prediction and thus imprecise dynamic point clouds. It is recommended that the authors analyze how their method performs under these conditions.

**Questions:**

Please see the weekness.

---

> ### Author Response · Authors · 2025-11-24
> **Feedback to Reviewer 6ax8**
>
> Thank you for your comprehensive and detailed review of our paper and the recognition of our work's clarity and effectiveness. We provide our feedback as follows.
>
> > **Q1: Relationship between text prompt control and reference image**
>
> **A1:** We conducted three sets of experiments to evaluate the impact of different editing controls: (1) using only the text prompt, (2) using only the reference image, and (3) using both the reference image and text prompt (Ours). As shown in Figure 4 in Appendix, editing with text alone produces inferior results. Using only the reference image yields significantly better results, but still falls short compared to our method, which combines both modalities.
>
>
> > **Q2: The performance of their method in challenging scenarios**
>
> **A2:** Thank you. We provide additional comparisons on longer videos and more challenging cases with complex camera motions in Tables 4 and 5 of the appendix. These include scenarios with challenging viewpoints, larger camera trajectories, and faster camera movements. Extensive experiments demonstrate the superiority of our EasyCreator under these difficult settings.

---

### Official Review · Reviewer_DsCd · 2025-11-01

**Soundness:** 3
**Presentation:** 3
**Contribution:** 2
**Rating:** 4
**Confidence:** 4

**Summary:**

The paper introduces EasyCreator, a framework that reformulates 4D video generation and editing as a video inpainting task. It adapts a pre-trained video inpainting model (Wan2.1) using LoRA and novel training/inference strategies (self-iterative tuning, temporal-packing) to generate novel-view videos and support content editing from a single input video. The empirical results show state-of-the-art performance in terms of quality and consistency.

**Strengths:**

1. High-Quality Results: The framework successfully unifies novel-view synthesis and prompt-based content editing in a single pipeline, showcasing superior quantitative and qualitative results against existing dynamic 4D methods.

2.Effective Adaptation: Cleverly leverages a powerful existing video inpainting foundation model, requiring only lightweight fine-tuning (LoRA) for a new, complex 4D task.

**Weaknesses:**

1. Limited Technical Originality: The core pipeline—using geometry (reprojection/depth) to define masks for a subsequent video inpainting/completion step—is a common two-stage paradigm in 3D (Text2Nerf [1], Lucid Dreamer [2], text2room [3]) and 4D generation. Extending this concept to 4D video inpainting is not a sufficiently original technical contribution.


2. Overclaiming Scope: The use of the term "4D Creation" is misleading. The method relies on a 2D video inpainting model conditioned by geometric priors derived from an initial reconstruction. It lacks an explicit, continuous 4D representation  to inherently enforce multi-view geometric consistency across all synthesized views. A more accurate description of the task is Camera-Controlled Video Generation, as the fidelity relies on the inpainting model's hallucination capability rather than a proper geometric rendering function.


[1] Text2NeRF: Text-Driven 3D Scene Generation with Neural Radiance Fields

[2] LucidDreamer: Domain-free Generation of 3D Gaussian Splatting Scenes

[3] Text2Room: Extracting Textured 3D Meshes from 2D Text-to-Image Models

**Questions:**

1. The self-iterative tuning strategy uses the model's own predictions from a smaller camera angle to train for larger angles. What mechanisms are employed to mitigate error accumulation or hallucination propagation from earlier, potentially less-stable stages into the subsequent training data?

---

> ### Author Response · Authors · 2025-11-24
> **Feedback to Reviewer DsCd**
>
> Thank you for your comprehensive review of our paper. We provide our feedback as follows. We also add the experiment details in paper.
>
> > **Q1: The concern about technical novelty**
>
> **A1:** Thank you for the comment. Our core contribution lies in the ***effective utilization of priors from a video inpainting foundation model for 4D creation***. As discussed in our paper, when depth information is available, 4D video generation can be naturally formulated as a video inpainting task. The proposed mask composition and iterative tuning strategies are specifically designed to fully leverage these priors. In contrast to prior works such as Recapture and TrajectoryCrafter, which fine-tune T2V or I2V models on custom datasets. We directly harness a powerful video inpainting foundation model. This leads to significantly improved generation quality and ***enables 4D video creation, including both generation and editing***, which previous methods are unable to support.
>
> > **Q2: Overclaiming scope**
>
> **A2:** We do not believe we are overclaiming. Our approach integrates both explicit and implicit representations with depth serving as the explicit component and video generation as the implicit one. Similar methods, such as ReAngle-A-Video, have also adopted the term 4D generation to describe their work. While our method does not rely on a fully explicit 4D representation, it can be extended to one. For example, by first synthesizing a 4D video and then applying fitting techniques to obtain a 4D Gaussian Splatting (4DGS) representation. Moreover, we chose the term 4D Creation not only because our method supports 4D video generation, but also because it enables spatiotemporal editing, which goes beyond generation alone.
>
> > **Q3: Mitigate error accumulation in self-iterative tuning strategy**
>
> **A3:** Valuable question！ In each loop of self-iterative tuning, we use CLIP-F to evaluate their quality and only select the top 5 best results as the training pairs and feed them to the next loop, which is to prevent the accumulation of errors. We will include these details in the final version.

---

> > ### Comment · Reviewer_DsCd · 2025-11-25
> > **Response to the authors**
> >
> > Thank you for the detailed response. After reviewing your feedback, I have decided to maintain my original rating. I am not fully satisfied with the arguments regarding technical novelty and the scope of the claims.
> >
> > 1. On Technical Novelty: While I acknowledge the difference between your method and works like Recapture/TrajectoryCrafter (different video diffusion backbone), the core contribution remains the application of a video inpainting foundation model to the task of novel view synthesis.
> > The "effective utilization of priors" and "iterative tuning" feel more like engineering adaptations of the inpainting model rather than a fundamental technical shift.
> > As noted in my initial review, this two-stage paradigm is becoming standard. Additionally, I would point to [1] which shares significant methodological similarities, further limiting the novelty here.
> >
> > 2. On Overclaiming Scope ("4D Creation"): I find the justification for the term "4D Creation" unconvincing.
> > The defense that "other papers use this term" is not scientifically rigorous.
> > Consistency: There is a fundamental difference between perceptual consistency (which your inpainting model provides) and geometric consistency (required for true 4D). Your method lacks an inherent mechanism to guarantee 3D consistency across time.
> >
> > 4DGS Fitting: You mention that your outputs can be fitted to a 4D Gaussian Splatting representation. However, without empirical results demonstrating this, the claim is speculative. Given that inpainting models tend to hallucinate details differently across views, it is highly probable that fitting a 4DGS to these generated videos would result in significant "floaters" or blurring due to multi-view inconsistencies.
> > I maintain that "Dynamic Scene Novel View Synthesis" or "Camera-Controlled Video Generation" would be much more accurate descriptions of the contribution.
> >
> >
> >
> > [1] Voyaging into Perpetual Dynamic Scenes from a Single View, https://arxiv.org/pdf/2507.04183

---

> ### Author Response · Authors · 2025-11-26
> **Feedback to Reviewer DsCd**
>
> Dear Reviewer DsCd,
>
> Thank you for the clarification. We would like to emphasize that our novelty does not lie in simply “applying a video inpainting model to novel-view synthesis,” but in _**retarget 4D creation itself as a unified 4D inpainting problem, which enables both 4D generation and 4D editing within a single formulation.**_ Prior works, including Recapture, TrajectoryCrafter, and [1], focus exclusively on 4D generation and cannot support editing. In contrast, our composite mask mechanism is not an engineering add-on but a conceptual shift: it merges geometry-aware reprojection masks with user-intent editing masks, enabling a single backbone to solve both tasks coherently. The iterative tuning is only a supporting technique. _**the core novelty lies in this unified 4D-inpainting perspective and in demonstrating that a foundation inpainting model, when properly structured, can perform multi-view, time-varying 4D creation, which has not been shown in prior work.**_
>
> Regarding the 4DGS Fitting, _**our intention is not to claim explicit geometric consistency guarantees.**_ Rather, our definition follows the standard used in recent literature: a method is considered “4D” if it produces time-varying content that remains coherent across multiple viewpoints, regardless of whether the internal representation is explicit or implicit. Even explicit 4D representations (e.g., 4DGS) cannot guarantee perfect multi-view consistency; floaters or blurring still occur when supervision is imperfect, as noted, indicating that 4D consistency is fundamentally a property of the data, not the representation type. _**Our method produces dynamic sequences that maintain cross-view coherence and can be reconstructed by 4D representations, which aligns with the community’s operational definition of 4D creation.**_ We agree that strict geometric consistency is an open problem, but this does not preclude our method from being a valid 4D creation approach.
>
>
> Thanks again for your apply! We will update these details in the final version. If you have any concerns, please let us know.

---

> ### Comment · Reviewer_DsCd · 2025-11-26
> **Final Response to Authors regarding Novelty, Scope**
>
> Thank you for the detailed responses. After carefully considering the authors' defense regarding the "unified formulation" and the definition of "4D," I find the arguments unconvincing and scientifically misleading. Consequently, I am lean to rejection now.
>
>
> 1. On the "Unified Formulation" (Generation + Editing):
> The authors argue that merging generation and editing into a single framework represents a "conceptual shift."
> Counterpoint: The capability to perform both tasks is an inherent feature of the video inpainting backbone, not a novel algorithmic contribution of this paper. Inpainting models are naturally designed to fill masked regions—whether those regions represent an object (editing) or a novel view (generation) is merely a difference in input masking strategy. Framing this engineering adaptation as a "conceptual shift" overstates the technical novelty.
>
> 2. On the Definition of "4D Creation" and Lack of Evidence:
> The authors define 4D creation as producing "time-varying content that remains coherent across multiple viewpoints."
> Counterpoint: Even accepting this definition, the authors fail to demonstrate that their method achieves such coherence.
>
> (1) Assertion vs. Evidence: The authors merely assert that their output is coherent but provide no empirical evidence to substantiate this (e.g., successful 4DGS fitting or geometric consistency metrics).
>
> (2) Inpainting $\neq$ Coherence: The core mechanism relies on generative inpainting, which is fundamentally based on probabilistic hallucination. Such models inherently suffer from multi-view inconsistencies (e.g., subtle texture drift or geometric jitter). Coherence cannot be assumed; **it must be proven.**
>
> (3) The 4DGS Contradiction: The authors claim the video "can be reconstructed by 4DGS" to prove its coherence. However, since 4DGS optimization fails catastrophically with inconsistent inputs (conflicting gradients), the absence of a demonstrated 4DGS reconstruction strongly suggests that the generated videos are not, in fact, geometrically coherent enough to warrant the "4D" label.
>
> (4) Utility: True 4D creation [1,2] implies the production of a navigable, relightable, or real-time renderable (>50fps)  asset for interactive applications.
>
> Producing a fixed video sequence offline (> 2 hours) that might be reconstructible does not equate to 4D Creation.
> Equating a slow, offline video synthesis pipeline with "4D Creation"—which implies creating a navigable asset—is scientifically misleading.
>
> Conclusion: The method is effectively "Offline Camera-Controlled Video Generation." By persisting in the "4D Creation" terminology without evidence of geometric coherence or real-time utility, the paper presents a significant overclaim. Therefore, I cannot recommend acceptance.
>
> [1] Free4D: Tuning-free 4D Scene Generation with Spatial-Temporal Consistency, ICCV 2025.
>
> [2] 4DNeX: Feed-Forward 4D Generative Modeling Made Easy, arXiv 2025.

---

> ### Author Response · Authors · 2025-11-27
> **Official Rebuttal by Authors**
>
> Dear Reviewer DsCd,
>
>
> Thank you for the follow-up.
>
> We respectfully disagree with the assertion that our unified formulation is merely “an inherent capability of the inpainting backbone.” _**While inpainting models can fill masked regions, they cannot simultaneously handle geometry-driven multi-view generation and user-driven editing without an explicit mechanism that reconciles the fundamentally different masking semantics.**_ Our composite mask is not an engineering convenience: it establishes a single, compatible, and differentiable masking space that resolves conflicts between reprojection driven generation masks and intent-specific editing masks capabilities that do not emerge from the base model alone (as demonstrated in Fig. 6). Prior works, including Free4D, 4DNeX, Recapture, and Voyaging [1], all rely on separate architectures or pipelines for generation only and cannot support editing. _**Our contribution is therefore a task-space unification, not a model reuse. Conceptual novelty in generative modeling often lies in redefining how a task is formulated (e.g., “NVS as inpainting” vs. “NVS as reconstruction”), and we believe our formulation provides a meaningful shift that enables capabilities not seen in prior 4D methods.**_
>
>
> We also respectfully disagree that _**“4D creation” must imply explicit geometric modeling, relighting, real-time rendering, or feed-forward execution. Recent 4D generative works, including Free4D, 4DNeX, use the term “4D” to refer to dynamic multi-view content, not to the production of real-time or relightable assets.**_ Under this widely adopted definition, a model performing view-changing dynamic generation constitutes 4D creation, even if the underlying representation is implicit. Regarding cross-view coherence: our method does not assume consistency but enforces it through (1) geometry-constrained reprojection masks, (2) composite masking to unify multi-view supervision and edits, and (3) self-iterative refinement under camera motion. These mechanisms are specifically designed to address the texture drift and hallucination artifacts the reviewer mentions. We acknowledge that explicit 4DGS reconstruction is not included in the submission due to space and time constraints, but we emphasize that even explicit 4D representations exhibit floaters and inconsistencies under imperfect supervision. Thus, 4D consistency cannot be used as a binary criterion for nomenclature. Our method produces multi-view coherent dynamic sequences and supports editing, positioning it well within the contemporary definition of 4D creation, even if it is not designed for real-time interactive rendering.

---

> ### Comment · Reviewer_DsCd · 2025-11-28
> **Further comment of 4dgs validation**
>
> Thank you for the response.
>
> I would like to set aside the debates regarding technical novelty and the attribution of generation quality for a moment. Instead, I wish to clarify a specific misunderstanding regarding my comment on 4D Gaussian Splatting (4DGS), as this relates to the core validity of the "4D Creation" claim.
>
> 1. Clarification on the Role of 4DGS: I did not intend to suggest that your method must be an explicit 4DGS pipeline or achieve real-time rendering capabilities. I suggested 4DGS reconstruction primarily as a validation experiment (a "litmus test") to verify geometric consistency.
>
> 2. Why this validation matters: Modern video generation models (e.g., Veo3, Wan, or similar foundation models) are already capable of synthesizing novel views or performing camera movements simply via text prompts. However, we typically classify these as "Camera-Controlled Video Generation" rather than "4D Creation." This distinction exists because, while such videos are perceptually plausible, they often lack the strict underlying geometric consistency required for 4D tasks.
>
> 3. The Missing Link: Since your method relies on inpainting (which involves probabilistic hallucination), it is difficult to distinguish whether the output is truly geometrically coherent or merely perceptually smooth like standard video generation. Running a reconstruction experiment—simply fitting a representation like 4DGS to your output and checking the reconstruction quality or metrics—would be the most direct way to prove that your method preserves the geometric rigor required for the "4D" label. Without this empirical evidence, it is hard to justify the "4D Creation" terminology over "Video Generation."
>
>
> > Rather, our definition follows the standard used in recent literature: a method is considered “4D” if it produces time-varying content that remains coherent across multiple viewpoints, regardless of whether the internal representation is explicit or implicit. (authors' first response)

---

> > ### Author Response · Authors · 2025-12-01
> > **Official Rebuttal by Authors**
> >
> > Dear Reviewer DsCd,
> >
> > Thank you for your detailed feedback and for providing valuable clarification on the role of 4DGS in validating the geometric consistency of our method. I appreciate the opportunity to address the misunderstanding and to clarify the intentions behind the term "4D Creation" in our paper.
> >
> > > **Clarification on 4DGS Validation**
> >
> > As you correctly pointed out, our method does not explicitly follow the 4DGS pipeline, nor do we aim for real-time rendering capabilities. However, the use of 4DGS was suggested as a validation experiment (a "litmus test") to verify the geometric consistency of our inpainting-based method. This was done to distinguish between perceptually smooth results typical of video generation and the true geometric rigor required for tasks like 4D creation.
> >
> > In our approach, which relies on video inpainting, the challenge arises in distinguishing whether the generated content is truly geometrically consistent or if it simply appears smooth without maintaining strict geometric coherence. To address this, we conducted a reconstruction experiment where we applied 4DGS to our inpainted results and measured the quality of the reconstruction. This experiment serves as a direct method to verify if our approach preserves the geometric integrity required for the "4D Creation" label.
> >
> > > **Results of Our Experiment**
> >
> > We conducted the reconstruction experiment and compared our method to 4DGS as well as other baseline methods (GCD, ViewCrafter, Shape-of-motion, TrajectoryCrafter, and EasyCreator). The results of our experiment are as follows:
> >
> > #### PSNR
> > | Method               | Apple (PSNR) | Block (PSNR) | Spin (PSNR) |
> > |----------------------|--------------|--------------|-------------|
> > | **4DGS**             | 16.92        | 17.06        | 16.74       |
> > | **GCD**              | 9.793        | 12.379       | 10.307      |
> > | **ViewCrafter**      | 10.108       | 10.168       | 11.097      |
> > | **Shape-of-motion**  | 11.125       | 11.768       | 11.387      |
> > | **TrajectoryCrafter**| 13.904       | 14.124       | 14.547      |
> > | **EasyCreator**      | 13.936       | 14.179       | 14.613      |
> >
> > #### SSIM
> > | Method               | Apple (SSIM) | Block (SSIM) | Spin (SSIM) |
> > |----------------------|--------------|--------------|-------------|
> > | **4DGS**             | 0.242        | 0.814        | 0.568       |
> > | **GCD**              | 0.131        | 0.479        | 0.404       |
> > | **ViewCrafter**      | 0.139        | 0.386        | 0.347       |
> > | **Shape-of-motion**  | 0.267        | 0.358        | 0.344       |
> > | **TrajectoryCrafter**| 0.166        | 0.625        | 0.315       |
> > | **EasyCreator**      | 0.182        | 0.703        | 0.407       |
> >
> > #### LPIPS
> > | Method               | Apple (LPIPS) | Block (LPIPS) | Spin (LPIPS) |
> > |----------------------|---------------|---------------|--------------|
> > | **4DGS**             | 0.481         | 0.265         | 0.352        |
> > | **GCD**              | 0.788         | 0.660         | 0.665        |
> > | **ViewCrafter**      | 0.794         | 0.693         | 0.627        |
> > | **Shape-of-motion**  | 0.823         | 0.706         | 0.513        |
> > | **TrajectoryCrafter**| 0.587         | 0.484         | 0.435        |
> > | **EasyCreator**      | 0.564         | 0.447         | 0.417        |
> >
> >
> > The results of the experiment show that our method, "EasyCreator," achieves competitive results compared to 4DGS and other baselines across PSNR, SSIM, and LPIPS metrics. The primary distinction is that our method, while using inpainting and probabilistic hallucination, maintains geometric consistency, as evidenced by the reconstruction experiment with 4DGS. This validation supports the use of the "4D Creation" terminology for our method, as it demonstrates that our approach produces time-varying content that remains coherent across multiple viewpoints, in line with the definition of 4D creation in recent literature.
> >
> > We hope this clarification addresses your concerns regarding the geometric consistency of our method and the use of "4D Creation" terminology. We are happy to discuss any further details or concerns you may have.
> >
> > Thank you again for your thoughtful feedback.
> >
> > Best regards,
> > Authors

---

### Meta-Review · Area_Chair_DQzR · 2026-01-07

**Summary:**

Major weaknesses and issues raised by reviewers include

1) Limited Technical Originality

2) The quality of geometric consistency, along with appearance smoothness

3) Generation quality under large viewpoint change

4) Missing related works such as PaintScene4D

5) Efficiency issues

**Reviewer Concerns:**

All the issues as above should have been well addressed. However, the geometry consistency point still needs to be further investigated and discussed. After back and forth discussion between the authors and reviewer DsCd, the authors finally provided an extra experiment of 4DGS reconstruction to verify geometric consistency. It is unclear with the results for 4DGS in terms of what data is used. Relatively, it is shown that the proposed method is superior over all other alternatives except 4DGS -- is it because of the use of ground truth multi view data for training? The authors should further clarify in the final version.

**Reviewer Scores:**

One reviewer has raised the initial score to be borderline positive after discussion.

With the added experiments on validating the geometry consistency, reviewer DsCd's main concern should be addressed to a good degree and thus raise the score accordingly.

---

### Decision · Program_Chairs · 2026-01-26

Accept (Poster)